# Sequence-specific dynamic DNA bending explains mitochondrial TFAM's dual role in DNA packaging and transcription initiation

Hyun Huh [1,5], Jiayu Shen [2,3,5], Yogeeshwar Ajjugal[3], Aparna Ramachandran[3], Smita S. Patel [3] ✉ & Sang-Hyuk Lee [1,4] ✉

Mitochondrial transcription factor A (TFAM) employs DNA bending to package mitochondrial DNA (mtDNA) into nucleoids and recruit mitochondrial RNA polymerase (POLRMT) at specific promoter sites, light strand promoter (LSP) and heavy strand promoter (HSP). Herein, we characterize the conformational dynamics of TFAM on promoter and non-promoter sequences using single-molecule fluorescence resonance energy transfer (smFRET) and single-molecule protein-induced fluorescence enhancement (smPIFE) methods. The DNA-TFAM complexes dynamically transition between partially and fully bent DNA conformational states. The bending/unbending transition rates and bending stability are DNA sequence-dependent—LSP forms the most stable fully bent complex and the non-specific sequence the least, which correlates with the lifetimes and affinities of TFAM with these DNA sequences. By quantifying the dynamic nature of the DNA-TFAM complexes, our study provides insights into how TFAM acts as a multifunctional protein through the DNA bending states to achieve sequence specificity and fidelity in mitochondrial transcription while performing mtDNA packaging.

Mitochondria are the cellular powerhouses that play a crucial role in ATP production through oxidative phosphorylation (OXPHOS). While ~77 OXPHOS proteins are nuclear-encoded and exported into mitochondria, mitochondrial DNA (mtDNA) encodes 13 OXPHOS proteins, 2 ribosomal RNA, and 22 tRNA in the mitochondria matrix[1]. In a human cell, the circular mtDNA is organized into mitochondrial nucleoids. Each nucleoid of ~100-nm diameter contains one or two copies of the ~16 kbp mtDNA, and ~1000 such nucleoids are distributed across the cellular mitochondria network[2]. Mitochondrial transcription factor A (TFAM), an essential protein[3], is a major component of the nucleoid responsible for compacting mtDNA through nonspecific binding and bending[4,5]. TFAM has two HMG-box (high mobility group box) domains, HMG-A and HMG-B, separated by a linker helix. Each HMG box comprises three alpha helices arranged in an L-shape that binds to

DNA minor groove and induce a nearly 90° DNA kink while the positively charged linker helix interacts with DNA minor groove to stabilize ~180° DNA bending by the two HMG-box domains[5–7].

In addition to mtDNA packaging, TFAM plays a crucial role in promoter selection and mitochondrial transcription initiation[8–12]. TFAM binds upstream of the two major promoters on the human mtDNA, the light strand promoter (LSP) and the heavy strand promoter (HSP), in a sequence-specific manner and recruits mitochondrial RNA polymerase (POLRMT) to form a preinitiation complex, which gets activated for transcription initiation upon binding to the mitochondrial transcription factor B2 (TFB2M)[10,12]. The crystal structure of the initiation complex shows that TFAM interacts with the tether helix of POLRMT via its HMG-B domain[10]. The TFAM-induced 180° U-turn in the upstream promoter DNA (−41 to −12) brings HMG-B close to

[1]Institute for Quantitative Biomedicine, Rutgers University, Piscataway, NJ 08854, USA. [2]Graduate School of Biomedical Sciences, Robert Wood Johnson Medical School, Rutgers University, Piscataway, NJ 08854, USA. [3]Department of Biochemistry and Molecular Biology, Robert Wood Johnson Medical School, Rutgers University, Piscataway, NJ 08854, USA. [4]Department of Physics and Astronomy, Rutgers University, Piscataway, NJ 08854, USA. [5]These authors contributed equally: Hyun Huh, Jiayu Shen. ✉e-mail: patelss@rutgers.edu; shlee@physics.rutgers.edu

POLRMT to make TFAM-POLRMT complex possible[10]. The TFAM C-terminal tail is vital in forming the distinctive U-shaped DNA[7,13,14]. The C-tail deleted TFAM can package mtDNA but is poor at supporting transcription from LSP and HSP[14,15].

Despite the insightful information, these crystallographic studies do not explain why only the TFAM molecules bound to specific promoter sequences exclusively initiate transcription. The crystal structures of TFAM bound to LSP, HSP, or nonspecific sequence (NS) reveal nearly identical bent structure of the DNAs[16]. However, the static crystal structures are susceptible to potential lattice artifacts and fail to uncover dynamic aspects of DNA-TFAM interactions expected under physiological conditions. TFAM-induced DNA bending in solution is demonstrated by ensemble biochemical and biophysical characterization[6,11,13,16], and some of these studies indicate the average bend angle is smaller for NS than LSP or HSP. Single-molecule fluorescence resonance energy transfer (smFRET) experiments on freely diffusing LSP-TFAM also reveal U-shaped DNA bending by TFAM[17]. Interestingly, molecular dynamics (MD) simulations suggest dynamic DNA bending/unbending transitions in the LSP-TFAM complex linked to folding/unfolding of the linker helix[17]; however, experimental evidence for dynamic bending/unbending has been lacking. A crucial question remains whether the dynamic nature of DNA bending is different in promoter and non-promoter DNA sequences.

In this work, we study the real-time dynamics of TFAM-induced DNA bending and unbending conformational changes at the single-molecule level using smFRET and single-molecule protein-induced fluorescence enhancement (smPIFE) assays, in combination with ensemble measurements. We find that the dynamic characteristics of the transitions between the two distinct DNA bending states, corresponding to partial (P) and full (F) bending of DNA, are highly sensitive to the DNA sequence; in particular, the LSP-TFAM complex is

exclusively stable in the F-state. The LSP-TFAM complex features a slower dissociation rate and smaller equilibrium dissociation constant than other DNA sequences. Based on these findings, we propose a model of interactions among DNA, TFAM, and POLRMT with the following primary features: (1) the partially bent DNA-TFAM conformational state performs DNA compaction; (2) the fully bent DNA-TFAM conformation is kinetically stable and prevented from dissociation; and 3) the fully bent DNA-TFAM conformational state mediates transcription initiation by the mitochondrial RNA polymerase. In this model, the stability of the fully bent conformation determines the kinetic stability and affinity of DNA-TFAM complexes in a sequence-dependent manner. This work provides insights into how the multifunctional protein TFAM and its dynamic interactions with mtDNA could mediate transcription with a high degree of sequence specificity while achieving mtDNA packaging in a sequence-nonspecific manner.

## Results
### TFAM bends LSP DNA at two bend angles dynamically
To monitor DNA conformational dynamics with smFRET, a 30 bp LSP sequence was end-labeled with Atto565 (donor) and Atto647N (acceptor) at −41 and −12 positions, respectively, and with (PEG-18)₂-biotin at the −41 end for surface immobilization (Fig. 1A, B). In the smFRET assay, ratiometric measurement of fluorescence from donor and acceptor that are attached to the ends of a DNA reports the real-time change in the end-to-end distance accompanied by DNA bending: the larger the bend angle, the shorter the end-to-end distance, resulting in brighter acceptor fluorescence and thus higher FRET value. The labeled DNA was sparsely attached to a passivated coverslip, and donor and acceptor fluorescence were simultaneously measured at 65 frame/sec using a homebuilt multi-channel-view total internal reflection fluorescence microscope (TIRFM)[18] under 561 nm excitation laser

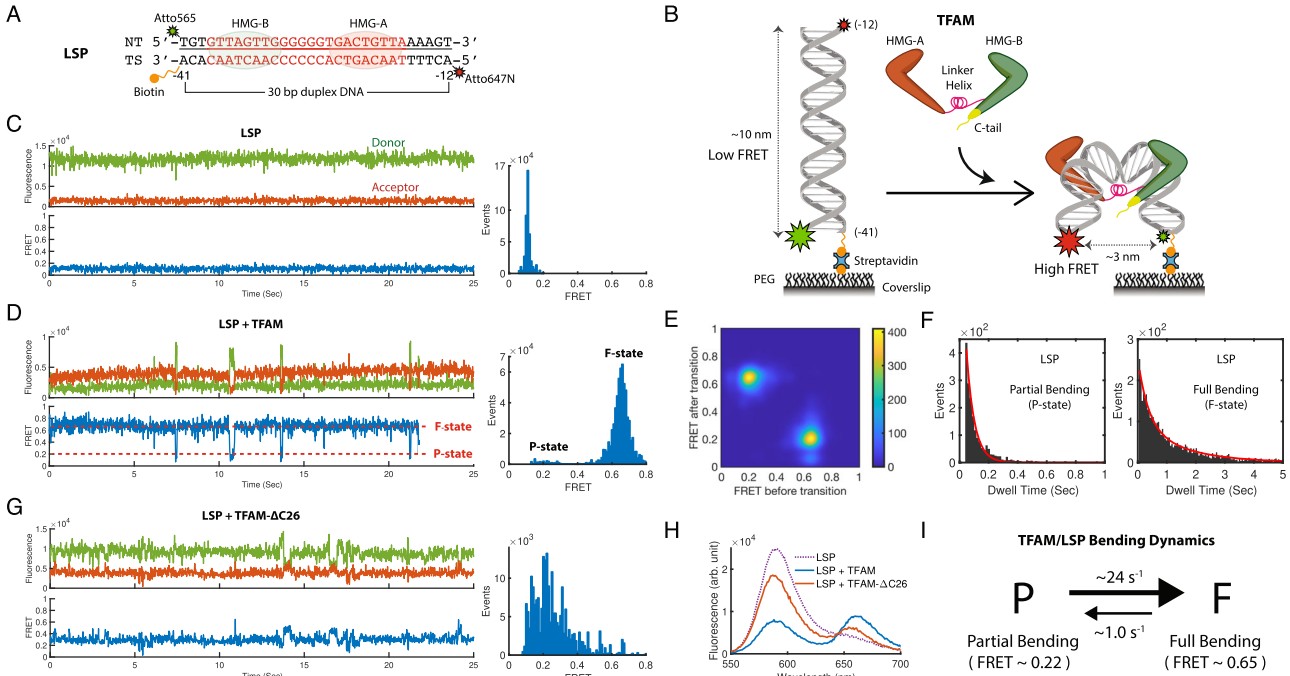

**Fig. 1 | TFAM bends LSP DNA in two bend angle states dynamically. A** LSP DNA labeled with a donor/acceptor dye pair for single-molecule FRET (smFRET) assay. **B** Schematic of smFRET assay to study real-time, single-molecule DNA bending dynamics of TFAM. **C, D, G** A sample smFRET time trace (left) and FRET histogram (right) for LSP DNA alone (N_Trace = 288), LSP + TFAM (N_Trace = 420), and LSP + TFAM C-tail mutant (ΔCT) (N_Trace = 219), respectively. **E** FRET transition density plot (TDP) for LSP + TFAM, showing two-state kinetics. **F** Dwell time histograms of the low FRET state, i.e., P-state (left), and the high FRET state, i.e., F-state (right). Red curves

indicate the fitted single- and double-exponential models for P- and F-state, respectively. **H** Ensemble FRET spectra, showing TFAM-induced DNA bending at the ensemble level in solution. **I** Diagram summarizes the two-state DNA bending kinetics of the LSP-TFAM complex. The forward and reverse rate constants (~24 s⁻¹ and ~1 s⁻¹) were calculated from the inverse of P-state and F-state mean dwell times (see Supplementary Table 1). smFRET time traces were fitted with Hidden Markov Model using vbFRET[20] or ebFRET[21] MATLAB software packages to plot FRET histograms, TDP, and dwell time histograms in this figure.

illumination (Methods). LSP alone showed steady fluorescence time traces with low acceptor and high donor signals, resulting in the FRET efficiency value of 0.11 (±0.02) (Fig. 1C). The low FRET value suggests a long end-to-end distance (~10 nm), representing the native unbent conformation of LSP DNA in the absence of TFAM.

The addition of TFAM dramatically increased FRET from 0.11 to 0.65 (±0.02), consistent with the U-shaped DNA conformation induced by TFAM (Fig. 1D, Supplementary Fig. 1). Although the smFRET time traces of LSP-TFAM were nearly stable around this high FRET value of 0.65, they occasionally transitioned to short-lived states with a lower FRET at 0.22 (±0.01). Hidden Markov model (HMM) fitting[19–21] and FRET transition density plot confirmed that the major transitions are occurring between the two FRET states (Fig. 1E). We designated the 0.22 and 0.65 FRET states as "P" and "F" to imply partial and full bending of DNA, respectively. A previous electron microscopy study[4] had revealed a broad range of DNA bend angles when TFAM non-specifically binds plasmid DNA, and MD simulation[17] hinted at the possibility of partial bending states. However, our smFRET data provide a definitive experimental evidence of partial DNA bending state and two-state dynamics between partial and full bending states in the LSP-TFAM complex. We quantified the two-state kinetics by fitting exponential models to the distribution of residence time (commonly termed 'dwell time') in each state (Fig. 1F, Supplementary Fig. 2), and obtained 1.0 (±0.3) sec and 0.04 (±0.01) sec as the average dwell times for the F- and P-state, respectively (Supplementary Table 1), showing a much higher stability in the F-state. The corresponding two-state kinetic model, summarized in Fig. 1I, shows a much faster forward rate (~24 s$^{-1}$) than the reverse rate (~1 s$^{-1}$).

TFAM mutant lacking the C-tail (TFAM-ΔC26) was previously shown to bend DNA, but not as well as wild type (WT)[13,15]. We conducted smFRET experiments with the mutant to characterize the role of C-tail in bending dynamics at the single-molecule level (Fig. 1G, Supplementary Fig. 3). The mutant could barely reach the full bending state; instead, it showed steady or less fluctuating smFRET time traces near 0.2 ~ 0.4 FRET values. Ensemble FRET measurement of LSP-TFAM confirmed the C-tail mutant's deficiency in bending DNA, consistent with the smFRET data (Fig. 1H). This suggests that C-tail is essential for stabilizing the F-state of the LSP-TFAM complex; without C-tail, the LSP-TFAM complex is kinetically trapped in the P-state with failed attempts to transition to the F-state. Structural studies of TFAM complexed with POLRMT and TFB2M[10] suggest that TFAM must be fully bent to interact with POLRMT. These interactions enable TFAM to recruit POLRMT and accurately position it over the transcription start site for de novo RNA synthesis. The lack of severe bending may explain why the C-tail deletion mutant of TFAM does not activate transcription[15].

## TFAM HMG-A and HMG-B stay bound to LSP DNA in the P-state

Intermediate macromolecular structures, such as the short-lived partial bending state of the LSP-TFAM complex observed in our smFRET assay, are not typically accessible by crystallographic methods. The MD simulation study of TFAM[17] suggested that the dynamic DNA bending can occur through reversible unfolding and refolding of the linker helix. The simulation indicated that the two HMG boxes remain bound to the minor groove during the DNA bending/unbending transitions (Model 1 in Fig. 2A). However, the P-state could also be created if one of the HMG-boxes, either HMG-A or HMG-B, dissociated from the DNA, releasing one of the ~90° kinks[5] (Model 2A or 2B in Fig. 2A). Studies suggest that HMG-B has weaker interactions with DNA than HMG-A[22]; hence, Model 2B could be considered more preferable than Model 2A.

To distinguish between the two mechanisms of DNA bending/unbending (i.e., Model 1 vs. Model 2), we used ensemble and single-molecule protein-induced fluorescence enhancement (smPIFE) experiments. smPIFE allows to monitor change in the proximity of a protein to a specific position on DNA where a PIFE-compatible dye (e.g., cyanine dye Cy3) is attached; because change in the local environment influences the cis-trans isomerization dynamics of cyanine dyes[23,24] and thus the fluorescence intensity. To test Model 1 and Model 2A, we labeled LSP with Cy3 at the −12 position, close to the HMG-A domain binding site[5–7,10] (Fig. 2A left panel). Studies above show that WT TFAM associates with LSP mostly in the F-state, whereas the C-tail mutant associates with LSP mostly in the P-state (Fig. 1D–G). If HMG-A is dissociated from the −12 arm in the P-state, the C-tail mutant will have a lower Cy3 PIFE than WT TFAM (Fig. 2A left panel). Surprisingly, the ensemble fluorescence spectrum showed the opposite result; the C-tail mutant had an increased Cy3 fluorescence intensity, whereas the Cy3 fluorescence increase by WT TFAM was considerably less (Fig. 2D). A control experiment with Atto565 labeled DNA at −41 did not show PIFE (Supplementary Fig. 4), validating the Cy3-specific PIFE phenomena. Because WT TFAM generates mostly the F-state and the C-tail mutant generates the P-state, we conclude that −12 Cy3 is closer to the HMG-A domain in the P-state than the F-state.

To further investigate the models at the single-molecule level, we performed smPIFE (Fig. 2B). LSP alone showed steady Cy3 fluorescence time traces typically between 1200 and 2000 in the unit of electron multiplying charge-coupled device (EMCCD) camera counts, with the histogram peaked at ~1600 (Fig. 2C left panel). The addition of WT TFAM increased this baseline fluorescence intensity between 2000 and 3000. More strikingly, the time traces showed occasional jumps to transient high PIFE states (Fig. 2B lower panel), much like the smFRET traces shown in Fig. 1D, but only a mirror image. The histogram showed a major Cy3 fluorescence peak at ~2300, corresponding to the baseline fluorescence, and a tail of higher counts from the short-lived fluorescence bursts (Fig. 2C right panel). In the case of the C-tail mutant, the smPIFE traces showed a dramatically increased baseline at around 4000, nearly double the LSP-alone value, as further confirmed by the Cy3 fluorescence histogram showing a broad distribution with a major peak at 4000 (Supplementary Fig. 4).

To correlate the PIFE signals to the FRET-based P-state and F-state, we simultaneously monitored PIFE and FRET using LSP labeled with Cy3 and Atto647N at −12 and −41 positions, respectively (Fig. 2E–G). Ensemble fluorescence spectrum of the dual-labeled LSP in the presence of WT TFAM reported a notable decrease in donor intensity and increase in acceptor intensity with nominal change in total intensity under 561 nm excitation, indicating a high FRET (i.e., F-state) without a significant PIFE enhancement (Fig. 2G). In contrast, C-tail mutant TFAM substantially increased both the donor and acceptor (i.e., large PIFE enhancement) while maintaining a low acceptor-to-donor intensity ratio (i.e., P-state). We then acquired single-molecule traces of Cy3 and Atto647N emissions to investigate the PIFE-FRET correlation from the surface-immobilized LSP under 561 nm excitation. Single-molecule traces of Cy3 and Atto647N labeled LSP showed a FRET behavior similar to the Atto565 and Atto647N labeled LSP case in the presence of TFAM, i.e., stable F-state of high FRET and transient P-state of low FRET (Fig. 2E lower left panels, Fig. 1D). Moreover, the single-molecule time traces showed higher levels of total fluorescence (i.e., the sum of Cy3 and Atto647N signals) in the presence than in the absence of TFAM, and it was more noticeably so for the low FRET state (i.e., P-state). 2-D histogram plots of Cy3 and Atto647N fluorescence constructed from many traces demonstrated this observation more clearly and definitively (Fig. 2E lower right panels).

The changes in Cy3 and Atto647N signals are induced by PIFE and FRET that affect the total and the relative intensity, respectively. Thus, PIFE and FRET can be evaluated from Cy3 and Atto647N signals. Decoupling the PIFE and FRET factors require calibration of fluorescence detection efficiency of donor and acceptor channels and their inter-channel crosstalk[25–27]. We used LSP labeled with Cy3 in the middle of the TFAM binding site on the DNA (between −20 and −21 positions) and Atto647N at −12 to determine the channel calibration parameters, which were applied to construct multivariate single-molecule time traces of PIFE and FRET for the Cy3 and Atto647N labeled LSP

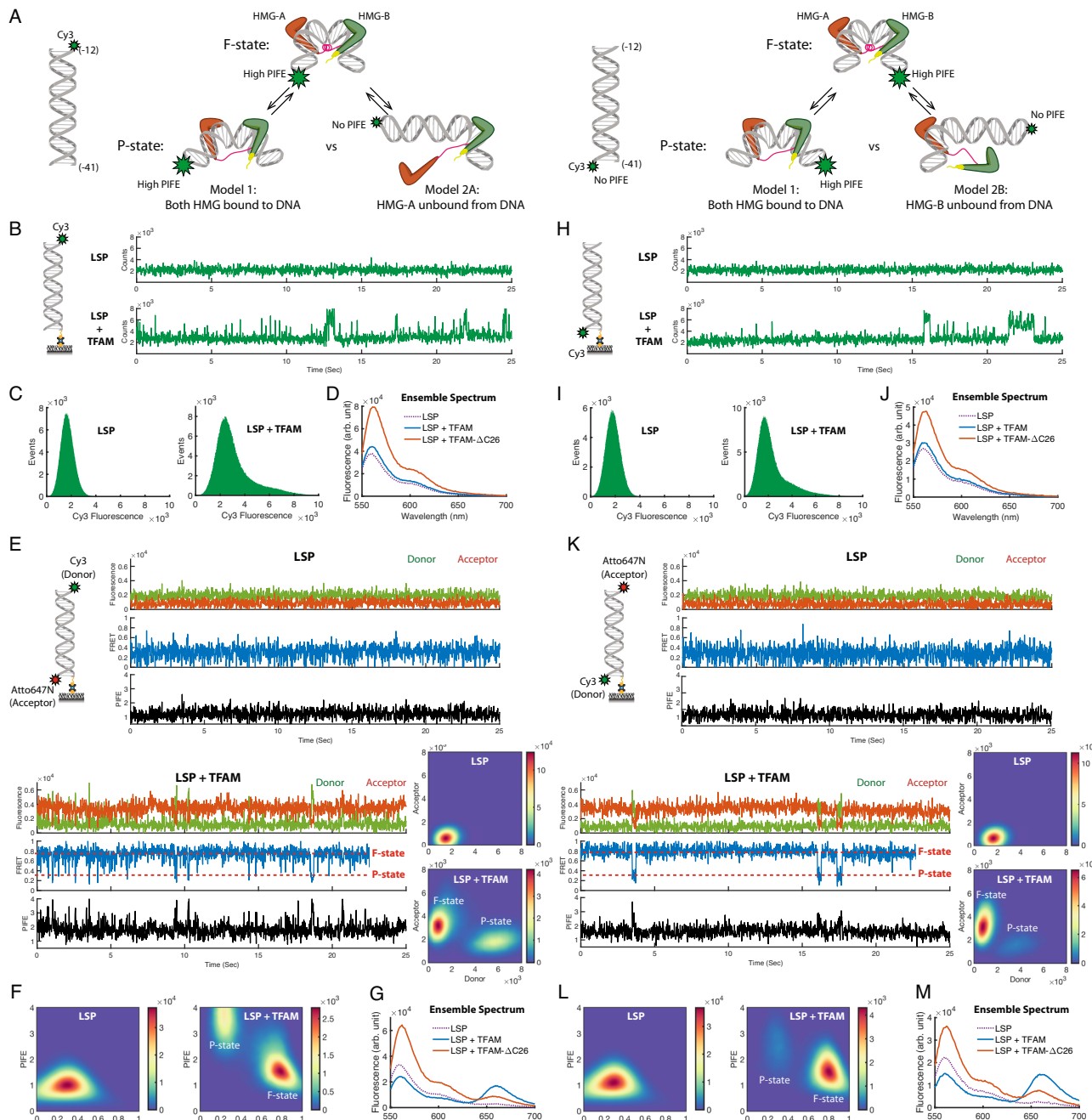

**Fig. 2 | [TFAM HMG-A and HMG-B stay bound to LSP DNA in the P-state].**
**A** Schematic of PIFE assay to test whether TFAM HMG-box domains stay bound to LSP DNA (Model 1) or dissociate from the DNA in the P-State (Model 2). Two subclasses of Model 2 were tested against Model 1 by switching the dye positions on the DNA: (**B**–**G**) for testing Model 2A with Cy3 labeled at −12 end; and (**H**–**M**) for testing Model 2B with Cy3 labeled at −41 end. **B** Sample single-molecule PIFE (smPIFE) time traces of LSP and LSP + TFAM. **C** smFRET histograms for LSP ($N_{Trace} = 200$), and LSP + TFAM ($N_{Trace} = 165$). **D** Ensemble Cy3 fluorescence spectrum, showing a strong PIFE effect when TFAM C-tail mutant is added to LSP DNA labeled with Cy3 at −12 end. **E** Hybrid smFRET/smPIFE assay using LSP DNA labeled with Cy3 (donor) and Atto647N (acceptor) at −12 and −41 end, respectively. Correlative donor/acceptor fluorescence enhancement due to PIFE effect (i.e., an overall increase in fluorescence intensity) upon addition of TFAM is shown, particularly for the P-state, from single-molecule time traces and 2-D histograms of

donor/acceptor fluorescence (bottom right: $N_{Trace} = 407$ for LSP and 79 for LSP + TFAM). PIFE and anti-correlative donor/acceptor fluorescence change due to FRET were decoupled using the procedure and method described in Methods. An example of reconstructed smFRET and smPIFE time traces is shown for LSP alone (upper panel) or when TFAM is added (lower panel). The PIFE time traces represent all the emitted photon numbers normalized to LSP DNA alone case. **F** 2-D histogram plot of FRET-PIFE values for LSP alone ($N_{Trace} = 407$) (left) and LSP + TFAM ($N_{Trace} = 79$) (right). Time traces and 2-D histogram show strong PIFE signals when LSP-TFAM dwells in the low FRET state (i.e., P-state). **G** Ensemble fluorescence spectra of Cy3/Atto647N-labeled LSP DNA show a strong PIFE effect and a moderate FRET effect simultaneously occurring with the addition of TFAM C-tail mutant. **H**–**M** −41 end-labeled Cy3 counterpart of **B**–**F**. $N_{Trace}$: 124 (LSP in **I**), 208 (LSP + TFAM in **I**), 417 (LSP in **K** and **L**), and 69 (LSP + TFAM in **K** and **L**).

(Methods, Supplementary Fig. 5). LSP alone showed steady time traces of FRET (at ~0.25) and PIFE (normalized to 1) (Fig. 2E upper panels). When TFAM was added, the stable high FRET F-state occasionally transitioned to the low FRET P-state. The smPIFE dynamics were anticorrelated with the smFRET dynamics, with stable baseline PIFE values at the F-state making occasional transitions to larger PIFE values at the P-state (Fig. 2E lower left panels). A 2-D histogram map of FRET and PIFE constructed from many single-molecule traces revealed an average PIFE value of ~1.5 and ~3.7 for the F- and P-state, respectively (Fig. 2F). Switching the Cy3 position to the opposite −12 end of LSP produced a similar result, except the average PIFE of the F-state was ~1.5 and P-state ~2.5 (Fig. 2H–M).

A strong PIFE effect was observed with Cy3 at the −12 or −41 end positions when the LSP-TFAM complex dwelled in the P-state, implying that both HMG-A and HMG-B are closer to the ends of LSP in the partially bent state than in the fully bent state. These data indicate that TFAM remains bound to LSP during the bending/unbending transitions, supporting Model 1. Moreover, our experiments indicate that the bending and unbending transitions move the two HMG domains laterally toward the DNA ends.

### HSP and NS follow the two-state DNA bending model, but the bending angles and kinetics differ from LSP

We next investigated how DNA sequence affects the TFAM-induced DNA bending dynamics, particularly on the human mitochondrial promoter, HSP, and a nonspecific sequence (NS) compared to LSP. A 30-bp DNA fragment with −12 to −41 HSP sequence and same length NS DNA of a random sequence were chemically synthesized and labeled with Atto565 and Atto647N fluorophores (upper panels of Fig. 3A, D). The smFRET time traces of HSP-TFAM showed a rapid dynamic

behavior, in contrast to the steady traces of LSP at the high FRET level (Fig. 3A lower left panels). The FRET histogram and transition density plot indicated that HSP-TFAM transitions between two states with FRET values of 0.27 (±0.01) and 0.51 (±0.01) (Fig. 3A lower right panel, Fig. 3B). Interestingly, these FRET values are not identical to those corresponding to the P-state and the F-state of LSP-TFAM shown in Fig. 1D, E. The low FRET state in HSP has a slightly higher value than LSP (0.27 vs. 0.22), and the high FRET state in HSP has a lower value than LSP (0.51 vs. 0.65). Dwell time distributions of the low and high FRET states fitted well to single-exponential functions with mean dwell times of 0.05 (±0.01) sec and 0.07 (±0.02) sec, respectively (Fig. 3C). Therefore, the high FRET state of HSP is ~14-fold less stable than LSP (0.07 s vs. 1.0 s), but the stabilities of the low FRET states in the two promoters are almost identical.

The smFRET traces of the NS-TFAM complex showed rapid two-state dynamics, similar to HSP-TFAM (Fig. 3D lower left panels). The low FRET state of NS was represented by a narrow peak at 0.21 (±0.01) FRET, similar to the P-state of LSP. However, the high FRET state 0.48 (±0.05) showed a more dispersed distribution around a lower mean value than LSP and HSP (Fig. 3D lower right panel, Fig. 3E). The mean dwell times from single-exponential model fitting were 0.12 (±0.03) sec and 0.07 (±0.01) sec for the low and high FRET states, respectively (Fig. 3F). Therefore, the low FRET state of NS is ~3-fold more stable than HSP (0.12 s vs 0.04 s), but the stability of the high FRET state is almost identical to HSP.

Figure 3G–K and Supplementary Table 1 summarize the bending and unbending kinetic parameters of the three DNA sequences. All DNA-TFAM complexes (LSP, HSP, and NS) form a partially bent P-state and a highly bent F-state, which dynamically interconvert on milliseconds time scale with distinct kinetics (Fig. 3G). The P-state has a

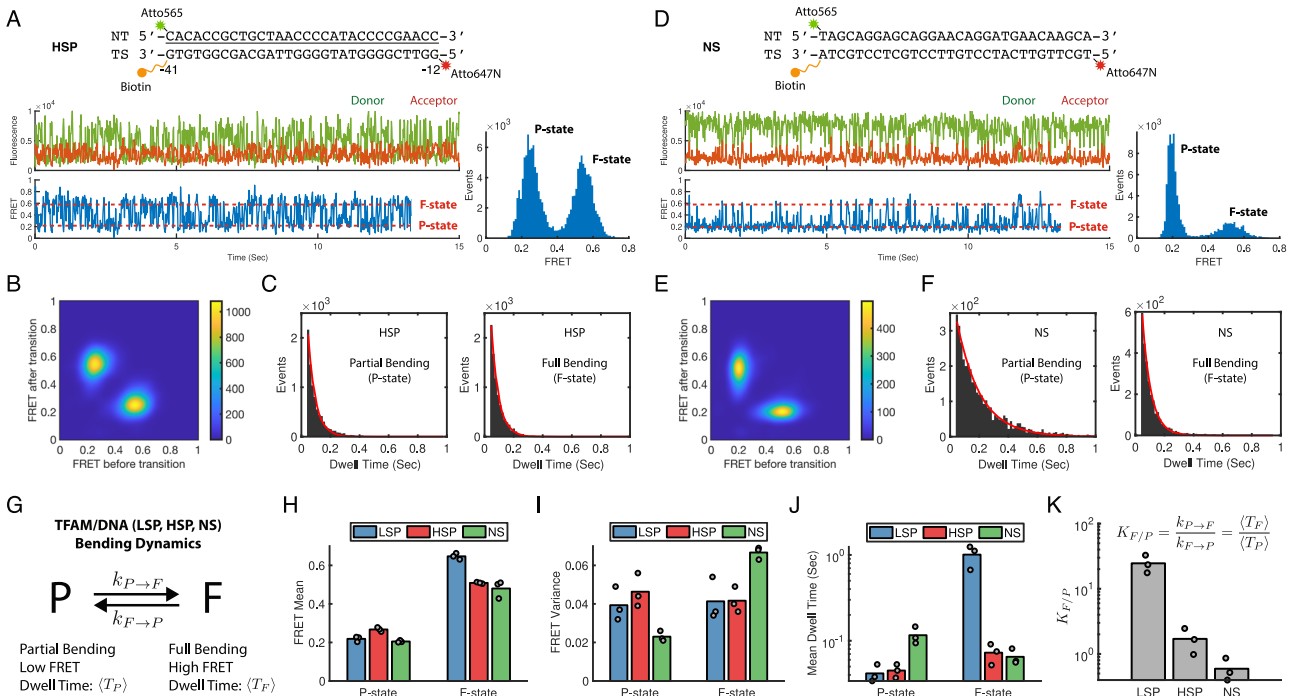

**Fig. 3 | [HSP and NS follow a two-state DNA bending model, but the bending angles and kinetics differ from LSP]. A–C** HSP-TFAM smFRET data. **A** Construct of Atto565/Atto647N-labeled HSP DNA, a sample smFRET trace, and FRET histogram ($N_{Trace}$ = 88). **B** FRET transition density plot. **C** Dwell time histogram and single-exponential model fit of P-state (left) and F-state (right). **D–F** NS-TFAM smFRET data ($N_{Trace}$ = 68) presented similarly to **A–C**. HSP and NS both show fast transitions between two FRET states. **G** Schematic of a unified TFAM-induced two-state DNA bending model, which is characterized by forward ($k_{P \to F}$) and backward ($k_{F \to P}$) rate constants, applies to all three DNA sequences. $\langle T_P \rangle$ and $\langle T_F \rangle$ represent mean dwell

time in P- and F-state, respectively. **H–K** Comparison of two-state (i.e, P- and F-state) kinetic model parameters of LSP, HSP, and NS DNA—data are represented as mean values that are obtained from $n$ = 3 distinct samples for each sequence. **H, I** Mean and variance of P- and F-state FRET values obtained from 2-D Gaussian mixture model fitting of the FRET transition density plots. **J** Mean dwell times of P- and F-state obtained from single-exponential model fitting of dwell time histograms (double-exponential model fitting for LSP F-state because of poor fit quality to single-exponential model. See Supplementary Fig. 2). **K** The equilibrium constant $K_{F/P}$ of the two DNA bending states.

similar mean FRET value in the three complexes, but the F-state mean FRET is higher in LSP than in HSP and NS (Fig. 3H). Notably, the NS complex has a higher F-state variance and a lower P-state variance than LSP and HSP (Fig. 3I). The different rates of bending ($k_{P \to F}$) and unbending ($k_{F \to P}$) in the three complexes (Fig. 3G) result in distinct dwell times of the P- and F-state (Fig. 3J). LSP and HSP have a shorter-lived P-state than NS, indicating that NS spends more time in the partially bent state than LSP and HSP. The F-state of LSP is strikingly long-lived than HSP and NS, indicating that LSP spends most of its time in the highly bent state. The bending/unbending equilibrium constant ($K_{F/P}$), calculated from the ratio of F-state and P-state dwell times, shows exceptional stability of the F-state for LSP (24.6) as compared to HSP (1.7) or NS (0.6); hence LSP and HSP show 42-fold and 3-fold better F-state stability than NS, respectively.

## TFAM dissociates slower and has higher affinity for LSP than HSP or NS

Our finding of the two conformational states led us to wonder whether TFAM's binding affinity for DNA is any different between P- and F-state, and if so, how it could result in a sequence dependence in the binding affinity. To answer this question, we studied DNA-TFAM association and dissociation for the three DNA sequences in detail. We first developed an ensemble FRET-based stopped-flow assay to measure the rate of TFAM dissociation from DNA. TFAM is incubated with Fluorescein isothiocyanate (FITC) and Tetramethylrhodamine (TAMRA) labeled DNA; a population of the dual-labeled DNA forming a complex with TFAM will bend to generate an ensemble FRET value at equilibrium; and rapidly flowing in a large amount of unlabeled DNA (trap DNA) will release the labeled DNA from TFAM, decreasing the ensemble FRET value (or equivalently acceptor signal) over time (Fig. 4A, Methods). TFAM dissociated from the dual-labeled HSP or NS within a minute, but it took over 30 min for the LSP (Fig. 4B). The normalized acceptor signal for HSP and NS followed single-exponential decay over time, and the fitted decay constant provided an off rate ($k_{off}$) of 6.7 min$^{-1}$ and 7.5 min$^{-1}$ for HSP and NS, respectively. The decay curve for LSP, however, followed a double-exponential function with two decay rates, 0.5 min$^{-1}$ and 0.065 min$^{-1}$. The slower rate presumably represented the slower time scale of the overall system equilibration due to the rebinding of the labeled LSP to TFAM, and therefore we interpreted the faster rate as representing the off rate for

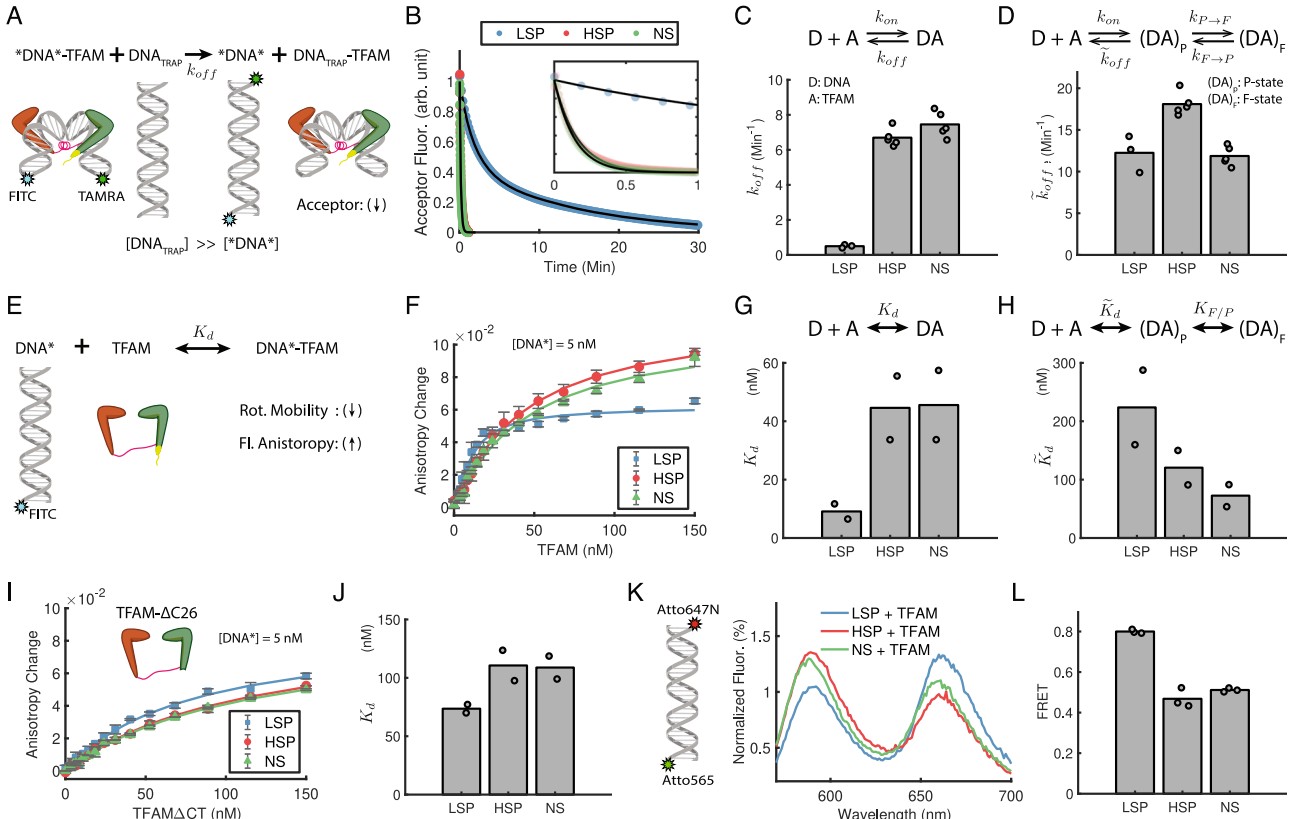

**Fig. 4 | [TFAM shows an exclusively slow dissociation rate and a strong affinity for LSP]. A–D** Ensemble FRET-based stopped-flow assay to measure the dissociation rate of the DNA-TFAM complex. **A** Schematic of the assay. *DNA* and DNA$_{TRAP}$ represent dual-labeled (with FITC and TAMRA) and unlabeled DNA, respectively. Adding a large amount of DNA$_{TRAP}$ causes release of *DNA* from TFAM, which is reflected in the FRET (or acceptor signal) change over time. **B** Acceptor fluorescence decay curves for the three DNA. The signals are normalized for better comparison among DNA sequences. Inset: magnified view to show the fast decay of HSP and NS curves within a minute. **C** The off rates ($k_{off}$) estimated by fitting the acceptor signal decay curves with exponential functions show an exclusively slow dissociation rate for LSP. **D** Upper panel: A kinetic model integrating DNA-TFAM binding/unbinding (with $k_{on}/\tilde{k}_{off}$ rate constants) and bending/unbending (with $k_{P \to F}/k_{F \to P}$ rate constants). Lower panel: The estimated pure off rate ($\tilde{k}_{off}$) of the P-state shows more moderate sequence dependence. **E–H** Ensemble fluorescence anisotropy-based assay to measure the equilibrium dissociation constant of the

DNA-TFAM complex. **E** Schematic of the assay. DNA* represent DNA labeled with FITC, and when it binds to TFAM, the fluorescence anisotropy increases. **F** Fluorescence anisotropy titration curves with increasing TFAM for the three DNA (5 nM). **G** The dissociation constants ($K_d$) estimated by fitting the titration curves with hyperbolic functions show distinctively tight binding of TFAM to LSP. **H** The two conformational states kinetic model results in the pure dissociation constant ($\tilde{K}_d$) with more moderate sequence dependence similar to the case of off rate. $K_{F/P}$ represents the equilibrium constant between the P- and F-state. Fluorescence anisotropy assays for C-tail mutant TFAM show the loss of sequence dependence in the titration curves (**I**) and the dissociation constants (**J**) upon C-tail deletion. **K, L** Ensemble FRET measurement of Atto565 and Atto647N labeled DNA (10 nM) upon addition of TFAM (50 nM). Also see Supplementary Fig. 6. Independent experiments were performed in duplicate (**F–J**) or triplicate (all the rest), and the data are represented by mean values.

LSP. The estimated off rate of TFAM for LSP was more than 10-fold slower than HSP or NS (Fig. 4C).

We noticed that this slower dissociation of TFAM from LSP correlated with the exclusively long dwell time in the F-state and its stability for the LSP-TFAM complex (Fig. 3J, K). The slow off rate can be explained by a two-step DNA-TFAM interaction model that involves the DNA bending conformational change after the bimolecular TFAM and DNA binding step (Fig. 4D upper panel). In this model, TFAM binding to DNA is weak in the P-state and strong in the F-state; TFAM binds to DNA to first form the weakly bound P-state with an on rate constant ($k_{on}$); the P-state can dissociate into TFAM and DNA with a pure off rate ($\tilde{k}_{off}$) or transform into the F-state with a bending rate ($k_{P \to F}$); and the F-state can transform back to the P-state without dissociation with an unbending rate ($k_{F \to P}$). Intuitively, the apparent off rate ($k_{off}$) measured from the stopped-flow experiment will always be slower than the pure off rate ($\tilde{k}_{off}$) because the DNA-TFAM complex is prevented from dissociation while dwelling in the F-state; hence, the more stable the F-state, the slower the apparent off rate. The bending and unbending rates (Fig. 3J) are at least an order of magnitude faster than the apparent off rate (Fig. 4C) for all the DNA. Under this condition, the P- and F-state establish a rapid equilibrium between them, and the apparent off rate is reduced from the pure off rate by the fractional occupancy of the P-state, i.e., $k_{off} = \tilde{k}_{off} / \left(1 + K_{F/P}\right)$ (Methods). The pure off rate ($\tilde{k}_{off}$), estimated from this relationship, showed more moderate sequence dependence than the apparent off rate ($k_{off}$) (Fig. 4C, D). Therefore, the observed sequence-dependence (especially the abnormally unique to LSP) that was present in the off rate measured from the stopped-flow experiment is explained in large part by the sequence-dependence of DNA bending/unbending dynamics of TFAM (i.e., $K_{F/P}$).

We next studied the sequence-dependence of equilibrium binding affinity between TFAM and DNA using an ensemble fluorescence anisotropy-based assay. The rotational mobility of the fluorescent molecule influences the fluorescence anisotropy value. An increase in the size of the molecule to which a fluorophore is attached leads to reduced rotational mobility, resulting in higher fluorescence anisotropy. Conversely, smaller molecules exhibit more free rotation, leading to a lower anisotropy value. The DNA-TFAM complex is much larger than DNA; hence, if we label DNA with FITC, we can measure DNA-protein interactions through the increased anisotropy of labeled DNA upon complex formation (Fig. 4E). Fluorescence anisotropy titration of FITC-labeled DNA with increasing TFAM showed hyperbolic binding curves from which we could determine the dissociation constant ($K_d$) of TFAM for the DNA (Fig. 4F, G). Our measurement shows that TFAM has ~5-fold tighter binding affinity to LSP ($K_d$ ~ 9 nM) than HSP ($K_d$ ~ 43 nM) or NS ($K_d$ ~ 44 nM). Depending on the buffer conditions and methods for $K_d$ measurements, studies have reported a 2-10-fold higher affinity of TFAM for LSP than HSP and NS[13,28]. This apparent dissociation constant $K_d$ is related to the pure dissociation constant $\tilde{K}_d$ in the two conformational states interaction model (Fig. 4H upper panel) in the same way as the case for off rate: $K_d = \tilde{K}_d / \left(1 + K_{F/P}\right)$ (Methods). The estimated pure dissociation constant showed more moderate sequence dependence than the apparent dissociation constant (Fig. 4H). Moreover, the discrimination for LSP in $K_d$ is lost upon C-tail deletion (Fig. 4I, J), which was observed earlier[22]. Interestingly, $\tilde{K}_d$ for WT TFAM and $K_d$ for the C-tail mutant are all very similar (~100 nM) for HSP and NS, and this may imply that the C-tail deletion mostly affects the bending/unbending kinetics of the DNA-TFAM complex with minor influences on the binding/unbinding of TFAM with DNA.

Lastly, we measured ensemble FRET of Atto565 and Atto647N labeled DNA upon the addition of TFAM (50 nM), which showed a noticeably larger FRET value for LSP (0.80) as compared to HSP (0.47)

and NS (0.51) (Fig. 4K, L, Methods). A larger ensemble FRET for LSP was observed earlier and interpreted as TFAM bending LSP at a greater bend angle than HSP or NS[6,11,13,16]. However, our dynamic two-state model alternatively explains it as rather the result of longer dwell time in the F-state for LSP (Fig. 3J, K).

## Discussion

Like histones in nucleosomes coating and packaging the genomic DNA, TFAM covers the human mtDNA, introducing local bends aiding DNA packaging. In addition to packaging mtDNA, TFAM is a transcription activator that recruits POLRMT to the two promoter sites forming a complex with TFB2M to catalyze efficient transcription initiation[29,30]. Because TFAM covers the mtDNA, it remains unclear how TFAM balances its DNA packaging and transcription activation functions. The two domains of TFAM, HMG-A and HMG-B, cooperatively bend DNA into a U-turn structure. In this work, we studied the DNA bending and unbending kinetics of TFAM on non-specific DNA and the LSP and HSP promoters using single-molecule FRET and PIFE methods. Our smFRET data revealed dynamic bending/unbending transitions of DNA-TFAM between two distinct conformational states, "F" and "P" states. The F-state is a severely bent state, likely representing the ~180° bent DNA conformation observed in DNA-TFAM crystal structure studies[6,7]. The P-state is a previously unreported intermediate state with a smaller DNA bend angle than the F-state. All three DNA-TFAM complexes (i.e., LSP, HSP, and NS) exhibited the two-state bending/unbending kinetics but with different transition rates that resulted in varying stability of the F-state (LSP > HSP > NS) (Fig. 3G–K). Interestingly, the stability of the F-state, as quantified by the equilibrium constant $K_{F/P}$ correlates directly with the dissociation rate and equilibrium binding affinity of TFAM for DNA (Fig. 4C–G). Our basic sequence analysis shows that the coexistent high-affinity TFAM-binding sequence motif and multiple sites of higher intrinsic DNA bendability are found only in LSP (Supplementary Note, Supplementary Fig. 7), which may provide clues to the sequence-dependent stability of the F-state. Interestingly, lowering salt (NaCl) concentration interrupts the F-state stability of LSP-TFAM complex (Supplementary Fig. 8), indicating salt-dependence of the aforementioned site-specific interactions and hence possible regulatory mechanisms of mitochondrial transcription.

Based on these findings, we propose a model explaining how TFAM regulates its sequence-specific mitochondrial transcription initiation on the LSP promoter site while achieving its sequence-nonspecific mtDNA packaging function (Fig. 5). In this model, TFAM initiates its association with DNA to generate a moderately bent DNA conformation (i.e., P-state). TFAM is weakly bound (in relative terms) to DNA in the P-state, possibly because the linker helix is unfolded partially and disrupts the DNA interactions with TFAM HMG-boxes and/or linker helix[17], and therefore the P-state dissociates into TFAM and DNA in ~6 s (Fig. 4D). Alternatively, the P-state can transition to a severely bent DNA conformation (i.e., F-state) and vice versa. The F-state likely represents the U-shaped DNA observed in crystal structure studies, and as such, TFAM is strongly bound to DNA in the F-state, particularly through the interaction between the folded linker helix and DNA minor groove; hence, it is unlikely for TFAM to dissociate directly from DNA in the F-state. Destabilization of the F-state, possibly through the linker helix unfolding and detachment from DNA, induces a transition to the P-state instead. The stability of the F-state is dependent on the DNA sequence, with LSP ~ 14-fold more stable than NS in the F-state, according to our smFRET data (Fig. 3J). Our proposition that the F-state is protected from dissociation explains the abnormally slow dissociation of TFAM from LSP. Besides, TFAM C-tail is critical to the formation of the F-state; C-tail deletion abolishes the transition to the F-state (Fig. 1G).

In our proposed model, both conformational states of DNA-TFAM can contribute to mtDNA packaging in principle. Our smFRET data indicate that the P-state has about ~2-fold longer residence time than

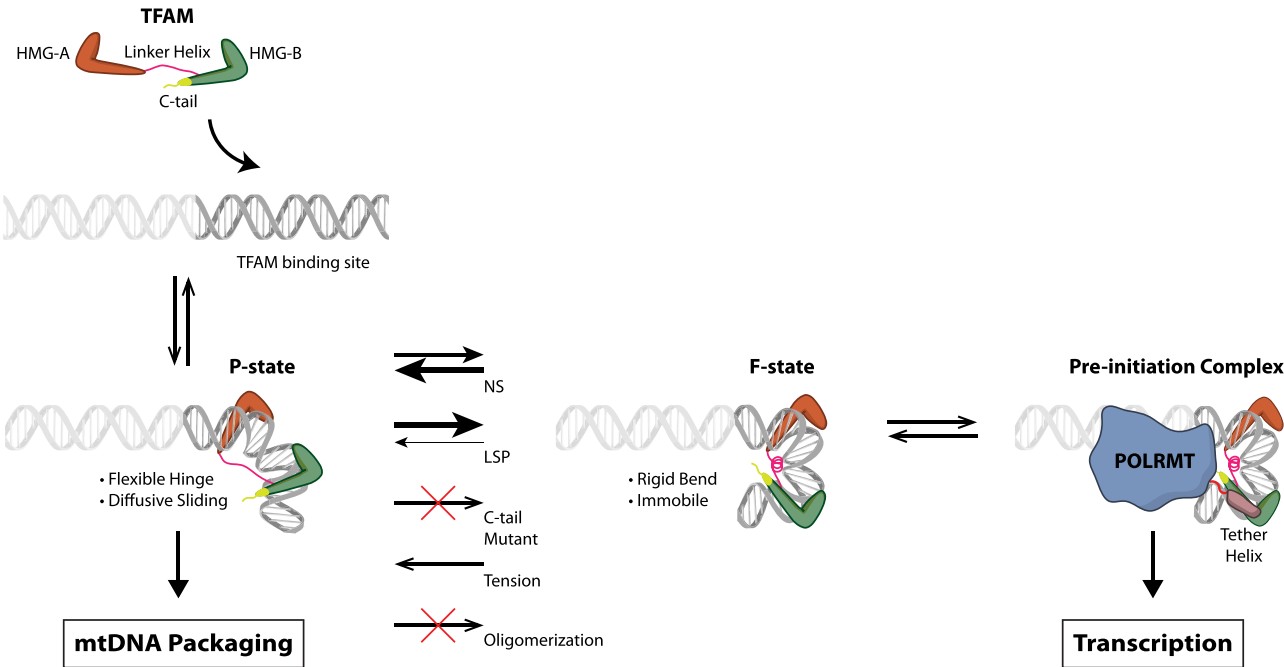

**Fig. 5 | [Model of how TFAM regulates its dual functions of mtDNA packaging and transcription initiation].** TFAM binds to DNA and forms a partially bent P-state with flexible hinges that support the diffusive sliding behavior on mtDNA, leading to TFAM oligomerization and patches that facilitate DNA packaging. The P-state dynamically transitions to the highly bent F-state, which facilitates POLRMT interaction and transcription activation at promoter sites. The F-state is less favorable on non-specific (NS) DNA and more favorable on LSP (promoter) DNA.

the F-state on NS DNA (Fig. 3J, K), implying TFAM molecules bound to non-specific regions of mtDNA will more often appear as a P-state than F-state. This is consistent with the previous electron microscopy study, which showed that a single mtDNA-bound TFAM largely induced less than 90° DNA bend in the non-promoter regions of DNA, and 180° U-turn bend was less frequently observed[4]. Therefore, the P-state is likely to play a more important role in mtDNA compaction. Other prominent features of TFAM, including DNA loop formation, diffusive translocation on mtDNA, and cooperative binding, were proposed to contribute to the mtDNA compaction[4,28,31].

A DNA loop would be generated if a pair of TFAM distally bound to DNA formed a dimer. Alternatively, a TFAM monomer could create a DNA loop if the two HMG-box domains slid on DNA in opposite directions. Our smPIFE data suggest that the two HMG-box domains slide toward the ends of DNA while transitioning from the F- to P-state (Fig. 2). The positively charged linker helix between the two HMG-box binds to the DNA minor groove to stabilize the F-state, and our studies suggest that the two domains stay bound to the DNA during the bending/unbending transitions. MD simulations of DNA-TFAM indicated that the linker unfolding contributes to DNA conformation dynamics[17]. We hypothesize that the helix unfolding detaches the linker from the DNA in the P-state, freeing the two HMG-boxes from the middle anchor point and allowing their translocation on DNA. Such events could result in 1-D diffusion of TFAM on the DNA and produce a DNA loop through the divergent motion component of the two HMG-boxes, as observed in the microscopy experiments[14].

Diffusive sliding of TFAM on DNA was observed earlier and was proposed as critical to the formation of highly stable DNA-TFAM patches through cooperative binding among multiple TFAM proteins and DNA[31]. It was unclear whether the sliding motion occurs in the P-state, F-state, or during inter-state transitions. The TFAM C-tail mutant defective in forming the F-state did not form DNA loops, partly because of impaired TFAM dimerization[22], but is nevertheless able to compact DNA[14]. Interestingly, ABF2, the yeast homolog of TFAM, involved in mtDNA packaging but not for transcription activation,

lacks the C-tail, which likely affects the stability of its fully bent state. A published structure of ABF2[32] shows that each HMG-box induces a 90° bend in the DNA. However, since this depiction is from a crystal structure, it might exclusively capture the stable fully bent state. In contrast, an AFM study[33] measured an average DNA bending angle of 78° from 43 DNA-TFAM complexes, encompassing various DNA bending angles. These results suggest that TFAM diffusively slides on DNA while in the P-state, and such translational motion is essential for stable compaction of mtDNA, whereas looping is not. It will be interesting to elucidate the potential correlation between DNA-TFAM conformational states and TFAM's translational mobility on DNA in future work with hybrid single-molecule methods such as Fleezers[34].

The U-shaped DNA observed in crystal structure studies of DNA-TFAM complex inspired a 'rigid bend' model suggesting that non-specific binding of TFAM to DNA generates ~180° bending and such 'rigid' bends of DNA underlie mtDNA compaction. However, previous studies with optical tweezers[31,35] challenged the 'rigid bend' model by showing that TFAM binding decreases the persistence length of Lambda DNA while slightly increasing the contour length, suggesting that the DNA becomes more flexible upon binding to TFAM. Additionally, the association and dissociation of TFAM from DNA was insensitive to tension, which is inconsistent with the rigid bend model[31]. A single HMG-box-A domain derived from human HMGB2 protein shares similar properties[35]. A 'flexible hinge' model was proposed to explain the mechanism of DNA compaction with increased flexibility induced by local denaturation of double-stranded DNA into single-stranded DNA upon TFAM or HMG-box binding. However, it was unclear how DNA-TFAM complexes could exhibit two seemingly incompatible behaviors: the rigid bend in the crystallographic U-turn DNA structure and the flexible hinge-like behavior in the optical tweezers assay. Our finding of the two conformational states, P- and F-state can help solve the puzzle. We propose that the P-state generates DNA denaturation and local flexibility in a force-insensitive manner, whereas the F-state imposes rigid bend angles in a force-sensitive manner; therefore, flexible hinges and rigid bends can

coexist. It can also be imagined that stretching DNA and increasing the tension may cause the F-state to transition to the P-state rather than TFAM to dissociate. Our smFRET data shows that the P-state is dominant in NS DNA (Fig. 3D–F), and thus the flexible hinge action of the P-state will subdue the force-sensitivity of TFAM dissociation in Lambda DNA or mtDNA. Furthermore, TFAM forms oligomers or patches on DNA[31], and the corresponding TFAM intermolecular interactions could further prevent transition to the F-state, forcing the mtDNA bound TFAM to reside in the P-state predominantly.

Unlike phage T7 RNAP, which binds to the promoter $\sim10^5$-order tighter than non-promoter sequences[36,37], the mtRNAPs from yeast[38] or human POLRMT[11], do not bind specifically to the promoter sequences and rely on transcription factors for specific initiation. Current models propose that TFAM recruits POLRMT to the transcription site to form a pre-initiation complex, which then binds to TFB2M to activate transcription initiation[1,10]. It seems essential that the TFAM bound to the promoter is more efficient at recruiting POLRMT to regulate high fidelity mitochondrial transcription, but the mechanism remains unclear. Structures of the pre-initiation complexes on LSP and HSP indicate that TFAM must be in a highly bent state to interact with the tether helix of POLRMT[10]. We propose that the F-state of the DNA-TFAM complex is more efficient at recruiting POLRMT and forming a mitochondrial transcription pre-initiation complex (Fig. 5), whereas the P-state and oligomers/patches of TFAM on non-specific DNA are more inert to POLRMT. The exclusive stability of the LSP-TFAM complex in the F-state observed in our smFRET data provides an explanation for how POLRMT can faithfully be activated by specific TFAM bound to the LSP promoter and construct a transcription complex in a sequence-dependent manner (Fig. 5). Our smFRET data show that TFAM can form the F-state on the NS DNA (Fig. 3). However, the F-state on NS is transient and has different characteristics than the F-state on LSP: lower mean FRET value and higher FRET variance for NS than LSP (Fig. 3H, I), suggesting that the F-state of NS may not be as efficient as the F-state of LSP in binding/activating POLRMT. While our model predicts the sequence specificity of transcription initiation at LSP, it does not readily explain how HSP achieves transcription initiation. TFAM bound to HSP is highly dynamic like on NS, but HSP has ~3-fold better F-state stability and a lower F-state FRET variance than NS (Fig. 3, 4). The cooperative binding of POLRMT to HSP-TFAM could assist in the stabilization of the F-state in HSP. Specific transcription on HSP and LSP is also achieved by sequence-specific interactions of POLRMT with the promoter initiation region[39]. Recent studies[14,39,40] showed higher in vitro transcription efficiency for LSP than HSP, reporting ~4- to ~10-fold higher amounts of transcription initiation transcripts from LSP than HSP; hence, these results are nevertheless consistent with the correlation between the stable bending of LSP and its higher transcription efficiency as proposed in our model.

Future work needs to determine the atomic structure of DNA-TFAM in the P-state, which has not been solved yet. Single-particle CryoEM techniques may be conducive to revealing the structures of transient states such as the P-state. The crystal structure of bending-deficient TFAM mutant, such as C-tail mutant, is also likely to provide insights into the structural conformation of the P-state. The 30-bp DNAs used in this work can only accommodate a TFAM monomer, and the results reported here represent the interactions between a TFAM monomer and a short DNA. Future work needs to investigate whether and how the characteristics of the DNA-TFAM conformational states/dynamics change as the DNA length increases, especially due to TFAM oligomerization. Several studies show that DNA-bound TFAM can dimerize using the surface of HMG-A[16] or the C-tail[14,22,41]. Therefore, the extension of our single-molecule study to longer DNA will be important in better understanding how TFAM interacts with the mtDNA in physiological conditions. In addition, the efficiency of POLRMT recruitment by DNA-TFAM complexes needs to be measured for promoter and nonspecific DNA sequences. Ensemble assays and the

technique of DNA curtains[42] combined with multicolor TIRFM imaging has great potential for high-throughput single-molecule localization of TFAM and POLRMT on a long DNA or even mtDNA to study the association among DNA, TFAM and POLRMT and its dependence on DNA sequence, protein concentration, etc. The hypothesis in our model that POLRMT is recruited by the F-state of the DNA-TFAM complex needs to be tested and explained. As a possible scenario, the TFAM HMG-B domain might be less ordered or sequestered in the P-state, disrupting the HMG-B interactions with POLRMT's tether-helix. The available structure of free POLRMT has not resolved the N-terminal extension in POLRMT harboring the tether-helix[43]. Single-particle CryoEM study has great potential in capturing the structures of interconverting intermediate states of the DNA-TFAM complex and interactions with POLRMT. Last but not least, future work also needs to address how transcription is activated at HSP but not at NS in spite of seemingly insignificant difference in DNA binding/unbinding and bending/unbending between HSP-TFAM and NS-TFAM complexes. A study[14] showed that randomizing the TFAM-DNA binding sequence in either LSP or HSP substantially decreased transcription activity, highlighting the essential role of the upstream promoter sequence for transcription initiation. A recent study[44] pointed out that TFAM interacts with the guanines in the $GN_{10}G$ motif in the TFAM DNA binding site, and this guanine pair is placed at an identical distance (−20/−31) from the transcription start sites in LSP and HSP. This suggests that TFAM must bend the DNA at a precise location upstream of the start site for optimal transcription initiation by POLRMT, which is not the case for NS. It was also reported that the basal transcription activity of POLRMT and TFB2M (without TFAM) is higher on HSP than on LSP[14], which may play an important complementary role in stabilizing the transcription initiation complex on HSP despite the less stable F-state in HSP-TFAM. Supporting this idea, our unpublished ensemble FRET data indeed showed that the addition of POLRMT and TFB2M compensated for the reduced bending ability of the C-tail mutant TFAM for both LSP and HSP, resulting in increased changes in FRET values. This complementary role of POLRMT and TFB2M in bending DNA was shown to rescue transcription activity even with the C-tail mutant TFAM[14]. Systematic smFRET studies on DNA-TFAM bending dynamics in the presence of POLRMT and/or TFB2M will provide further clues to the puzzle related to HSP.

## Methods

### Expression and purification of TFAM (43–245) and TFAM-ΔC26 (C-tail deletion mutant) proteins

For recombinant protein expression and purification, *TFAM* (43–245, lacks C246) were cloned into *Escherichia coli* expression vector pPRO-EX-HTb, which encodes a cuttable N-terminal 6xHis-tag by Tobacco Etch Virus (TEV) protease. The ΔC26 (43-220) TFAM C-tail deletion mutation was expressed as SUMO-fusion protein, which was cleaved during the purification (kindly provided by Dr. Craig Cameron).

TFAM (43–245) was expressed in *E. coli* strain BL21 codon plus (RIL) (AGILENT TECHNOLOGIE) and purified through two column chromatographic steps: $Ni^{2+}$ Sepharose 6 HP followed by Heparin Sepharose (Cytiva). The cells were harvested, resuspended, and lysed in lysis buffer (50 mM Na-phosphate (pH 7.8), 300 mM NaCl, 10% glycerol, 20 mM imidazole, and protease inhibitors (Roche)). The clarified lysate was subjected to two-column chromatography steps: $Ni^{2+}$ Sepharose 6 HP followed by Heparin Sepharose (Cytiva). The purified TFAM was treated with His-tagged TEV protease in 1:50 ratio to cleave the His-tag. The His-tag, uncut proteins, and His-tagged TEV protease were removed by reverse Ni-NTA column. Purified TFAM was concentrated using 10 kDa MW centrifugal filter units (MilliporeSigma) and stored at −80 °C.

TFAM-ΔC26 (C-tail deletion) was expressed in *E. coli* strain Rosetta (DE3) (Novagen). Cells were grown at 37 °C shaker in LB (Luria-Bertani) liquid media broth containing 50 and 25 μg/mL Kanamycin and

Chloramphenicol until an optical density (OD) at 600 nm reached 0.8. The induction of recombinant proteins was conducted with 1 mM Isopropyl β-D-1-thiogalactopyranoside (IPTG, EMD Millipore Corp.) overnight at 16 °C shaking at 220 rpm.

The cells were pelleted, resuspended, and lysed in lysis buffer (100 mM Na-phosphate (pH 8.0), 500 mM NaCl, 20% glycerol, 5 mM imidazole, 0.1% NP40, 10 mM β-mercaptoethanol, 0.25 mM EDTA and protease inhibitors (Roche)). The clarified lysate was subjected to Ni$^{2+}$ Sepharose 6 HP (Cytiva). Ulp1 protease (1 μg per 1−5 mg SUMO fusion) was added to the collected Ni column elution to cleave the Sumo-tag. Duration of cleavage was overnight at 4 °C during dialysis in dialysis buffer (50 mM Na-phosphate (pH 8.0), 200 mM NaCl, 10% glycerol, 1 mM EDTA, 1 mM DTT and protease inhibitors (Roche)). The protein was purified with reverse Ni-NTA and heparin column to remove Sumo-tag, uncut proteins, Ulp1 protease, and other contaminants. Purified TFAM was concentrated using 10 kDa MW centrifugal filter units (MilliporeSigma) and stored at −80 °C.

## Fluorophore labeled oligonucleotides

LSP, HSP, and NS DNA were constructed from complementary single-stranded non-template and template oligonucleotides of the following sequences:

LSP (non-template 5′-TGTGTTAGTTGGGGGGTGACTGTTAAAAGT-3′,

template 5′-ACTTTTAACAGTCACCCCCCAACTAACACA-3′);

HSP (non-template 5′-CACACCGCTGCTAACCCCATACCCCGAACC-3′,

template 5′-GGTTCGGGGTATGGGGTTAGCAGCGGTGTG-3′); and

NS (non-template 5′-TAGCAGGAGCAGGAACAGGATGAACAAGCA-3′,

template 5′-TGCTTGTTCATCCTGTTCCTGCTCCTGCTA-3′.

The NS sequence is adopted from a previously published paper[13] and originates from Mus musculus mitochondrion (NCBI Sequence ID: NC_005089.1, from coordination 5684 to 5708). All DNA oligonucleotides were custom synthesized and purified by high-performance liquid chromatography (Integrated DNA Technologies, Coralville, IA, USA). For FRET studies, Atto565 was introduced at the 5′-end of the non-template and Atto647N at the 5′ end of the template strand. For PIFE studies, Cy3 was labeled at either the 5′ end of the template or non-template strands. For hybrid PIFE-FRET studies, Cy3 and Atto647N were labeled at 5′ non-template and 5′ template strands, respectively, or vice versa. For calibration of donor and acceptor channel detection efficiency as needed for PIFE-FRET assay, Cy3 was labeled at between 9 and 10th nucleotide position from the 3′ non-template end of LSP DNA while Atto647N was labeled at the 5′ template end. For fluorescence anisotropy measurements to study DNA-protein interaction, DNA was labeled with Fluorescein at the 3′ end of template strand. In case of single-molecule assays, biotin-2×PEG18 was introduced at the 3′ end of the template for immobilization on the surface. DNA duplex constructs were generated by annealing equal moles of single-stranded non-template and template oligonucleotides. Typically, 100 μM stock solutions of complementary single-stranded DNA constructs were mixed and 10× diluted with a buffer (20 mM Tris, pH 8.0, 100 mM NaCl, 1 mM EDTA), incubated at 95 °C for 2 min, and slowly cooled down to the room temperature for several hours.

## Fluorescence anisotropy-based titrations to measure protein-DNA K$_D$ values

Fluorescence anisotropy was measured on Spark® Cyto (Tecan Trading AG, Switzerland). Fluorescein labeled −41 to −12 LSP/HSP/NS DNA (5 nM) was mixed with 0−75 nM serially diluted TFAM, and the excitation and emission wavelength were set at 480 nm (15-nm bandwidth) and 530 nm (20-nm bandwidth), respectively. Experiments were conducted in reaction buffer (50 mM Tris acetate (pH 7.5), 100 mM sodium glutamate, 50 mM NaCl, 10 mM magnesium acetate, 0.5 mM

TCEP, and 0.01% Tween 20, 0.1 mg/ml BSA) at 25 °C in a 384-well plate (Tecan Trading AG, Switzerland). Fluorescence anisotropy values were calculated automatically in the Software provided by the equipment.

## Stopped-flow kinetics to measure the off rate

FRET-based stopped-flow assays were utilized to measure TFAM dissociation rate constants from DNA at 25 °C. Syringe A of the stopped-flow instrument (Auto-SF 120, Kintek Corp, Austin, Tx) contained a mixture of Fluorescein and TAMRA dual-labeled DNA (20 nM of −41 to −12 LSP, HSP, or NS DNA) and TFAM (60 nM) in reaction buffer (50 mM Tris acetate, pH 7.5, 100 mM Na-glutamate, 50 mM NaCl, 10 mM magnesium acetate, and 0.5 mM TCEP), while syringe B held the DNA trap (unlabeled 30 bp NS DNA, 600 nM). The acceptor TAMRA fluorescence emission was measured using a 570 nm cut-off filter after excitation at 478 nm. The resulting change in fluorescence intensity over time was fitted to estimate the off rates.

## Ensemble Förster Resonance Energy Transfer (FRET) measurements

Ensemble FRET measurements were performed on Fluoromax-4 spectrofluorometer (HORIBA). Experiments were conducted in transcription reaction buffer (50 mM Tris acetate (pH 7.5), 100 mM sodium glutamate, 10 mM magnesium acetate, 10 mM DTT, and 0.01% Tween 20, 0.1 mg/ml BSA) at 25 °C. Atto565-Atto647N dual labeled −41 to −12 LSP/HSP/NS DNA (10 nM) was incubated with TFAM (50 nM) for 5 min. The fluorescence intensity ($F_{DA}$) in the 570-700 nm spectral range was measured with excitation at 561 nm ($\lambda_{EX}^D$). The fluorescence intensity of Atto565 single labeled DNA ($F_D$) was measured in the same way. The acceptor fluorescence ($F_A$) of dual labeled DNA upon direct excitation at 649 nm ($\lambda_{EX}^A$) in the 655−730 nm range was also determined. The FRET efficiency was calculated using the (ratio)$_A$ method[13,36,45]. The acceptor fluorescence due to FRET ($F_{DA}$) was divided by the acceptor fluorescence with direct excitation ($F_A$) to determine the (ratio)$_A$:

$$(ratio)_A = \frac{F_{DA} - N * F_D}{F_A},\qquad(1)$$

where $N \equiv F_{DA}\left(\lambda_{EX}^D, \lambda_{EM}^D\right)/F_D\left(\lambda_{EX}^D, \lambda_{EM}^D\right)$ is the normalization factor of donor fluorescence in the presence and absence of FRET; $\lambda_{EX}^D$ is the donor excitation wavelength; and $\lambda_{EM}^D$ is the wavelength at donor emission peak 589 nm. FRET efficiency ($E_{FRET}$) was calculated using its mathematical relationship with (ratio)$_A$:

$$(ratio)_A = E_{FRET}\frac{\epsilon_D\left(\lambda_{EX}^D\right)}{\epsilon_A\left(\lambda_{EX}^A\right)} + \frac{\epsilon_A\left(\lambda_{EX}^D\right)}{\epsilon_A\left(\lambda_{EX}^A\right)},\qquad(2)$$

where $\epsilon_D(\lambda_{EX}^D)$ is the extinction coefficient of the donor (Atto565) at its excitation wavelength ($\lambda_{EX}^D$); and $\epsilon_A(\lambda_{EX}^A)$ and $\epsilon_A(\lambda_{EX}^D)$ are the extinction coefficients of acceptor (Atto647N) at its excitation wavelength ($\lambda_{EX}^A$) and at donor's excitation wavelength ($\lambda_{EX}^D$), respectively. The $\frac{\epsilon_A(\lambda_{EX}^D)}{\epsilon_A(\lambda_{EX}^A)}$ value of 0.066 and the $\frac{\epsilon_D(\lambda_{EX}^D)}{\epsilon_A(\lambda_{EX}^A)}$ value of 0.65 were used.

## Microscope instrument and single-molecule data acquisition

We acquired single-molecule fluorescence (i.e, smFRET and smPIFE) data using a homebuilt total internal reflection fluorescence microscope (TIRFM) that is based on a commercial microscope (Nikon, Ti-E) and is equipped with multiple lasers (405 nm, 488 nm, 561 nm, 637 nm), an electron multiplying charged-coupled device (EMCCD) camera (Andor, iXon Ultra-888), and a custom split-camera-view system for simultaneous donor and acceptor fluorescence imaging[18]. Prior to the actual experiment, we imaged 100 nm multispectral

fluorescence beads (Invitrogen, TetraSpek) in both the donor and the acceptor fluorescence channels of the split-camera-view system in order to align the donor and the acceptor imaging channels each other through a post-processing. For smFRET and smPIFE experiments, a sample chamber was constructed with a #1.5 coverslip that was coated with PEG (PEG:biotin-PEG = 1:20) to prevent nonspecific binding processes. To immobilize dsDNA on the coverslip surface, a streptavidin solution (0.1 μg/ml with 0.1% BSA) and a 10 nM biotinylated dsDNA sample solution were sequentially injected to the chamber, and the untargeted components in solution were removed by washing buffer (20 mM Tris, pH 8.0, 50 mM NaCl) after incubation for 10 minutes. TFAM mixed with an imaging buffer (20 mM Tris, pH 8.0, 50 mM NaCl, 2 mM EDTA, Trolox, 1% glucose, GODCAT, 10 mM MgCl2) was injected and incubated for 10 minutes prior to the data acquisition. We used 561 nm laser (Coherent, MX561-1000 STM) to excite the donors (Atto565 or Cy3) and typically acquired 4000 image frames with an exposure time of 15 msec in Nikon's ND2 file format using Nikon NIS-Elements software.

## Single-molecule FRET data analysis

All the single-molecule analysis was performed using a custom developed MATLAB software other than Hidden Markov Model (HMM) fitting of smFRET traces that was done using publicly available vbFRET and ebFRET packages. The data analysis procedure is as follows. First, donor and acceptor images were extracted from a time-lapse stack of split-camera-view images, and they were aligned to each other using an affine transformation that was predetermined by the calibration images of 100 nm multispectral fluorescence beads. Second, a subset (e.g., 10%) of the two-channel-aligned time-lapse image stack was averaged; the contrast of the averaged image was improved by Laplacian of Gaussian spatial filtering; the aligned donor and the acceptor images were summed; bright spots in the donor-acceptor-summed image were identified as candidates of single molecules using intensity thresholding (typically the center peak intensity of a selected bright spot should be more than 8-fold brighter than the background intensity fluctuation level); 11 × 11 pixel images around the local peak intensity pixels were cropped and fitted with 2-D Gaussian functions for sub-pixel precision localization; and outliers found from the spot size/brightness distribution were discarded. Third, using herein found locations as a reference, the bright spots were tracked in time to estimate the drift motion (the cropped time series images were downsampled by averaging every 10 images to improve the signal-to-noise ratio and localization precision, and the down-sampled drift data was extrapolated to provide the drift value for every time) that was used to extract the drift-corrected single-molecule donor and acceptor time series. Fourth, donor-acceptor-summed time traces were fitted with HMM to identify/remove donor-bleached segments; and smFRET time traces were also fitted with HMM to identify/remove TFAM-unbound DNA and acceptor-bleached segments. Fifth, the HMM-fitted smFRET traces were further analyzed to generate FRET distributions, TDP plots, dwell time distributions, etc.

smFRET studies involve a series of complex computational analysis steps to infer the number of intrinsic molecular states and the transition rates among them from single-molecule time traces. Individual components of smFRET analysis are still active fields of research and there exist a growing number of analysis methods and tools available. However, any statistical method to predetermine a sufficient number of traces and data points for a good estimation of the molecular properties is not well established yet. A recent benchmark study[46] on smFRET analysis demonstrated that simulated 75 traces (59,486 data points) and experimental 19 traces (226,100 data points) corresponding to two-state models led to very similar results among 14 different analyses performed by multiple laboratories. In our smFRET analysis presented in Fig. 1D–F and Fig. 3A–F, we used 420 traces (648,050 data points) for LSP, 88 traces (143,988 data points) for HSP,

and 68 traces for NS (89,475 data points). The number of traces and data points for LSP were chosen especially larger than HSP and NS cases because of more than 5-fold slower transition kinetics for LSP. Our smFRET data sizes are comparable to the values used in the previous benchmark study and typical smFRET studies. Moreover, the appearance of two clear populations in the FRET distributions and the very good exponential model fitting results of the dwell time distributions as shown in the figures justify that the sample sizes used in our study are sufficient for reasonable estimation of the relevant parameters regarding the DNA-TFAM conformational dynamics. In addition, we repeated independent experiments with three distinct samples for each DNA sequence to test reproducibility. We incorporated the 95% confidence intervals from the model fitting and the variations in the triplicate data sets into the final uncertainty in the parameter estimates as presented in Supplementary Table 1.

## Single-molecule hybrid PIFE-FRET data analysis

We consider a situation in which a pair of FRET donor and acceptor fluorophores is illuminated by donor-excitation light. If the two fluorophores are far away from each other such that there is no FRET, the ideal donor and acceptor intensity (i.e., photons per unit time emitted from a single donor and a single acceptor) $D_0$ and $A_0$ is given by

$$D_0 = Q_D U = q_D u \bar{Q}_D \bar{U}, A_0 = Q_D U = q_A u \bar{Q}_A \bar{U}, \qquad (3)$$

where $U$ is the excitation light power, and $Q_D$ is the quantum efficiency of donor emission by the donor-excitation light, and $Q_A$ is the quantum efficiency of acceptor emission by the donor-excitation light. $U$, $Q_D$, and $Q_A$ are each normalized by reference values $\bar{U}$, $\bar{Q}_D$, and $\bar{Q}_A$, respectively, and are represented by $u$, $q_D$, and $q_A$. If the donor and acceptor are close, their emission intensities $D$ and $A$ depend on the FRET efficiency $f$ according to

$$D = (1-f)D_0, A = fD_0 + A_0 \qquad (4)$$

Introducing a matrix $E = \begin{pmatrix} e_{DD} & e_{AA} \\ e_{AD} & e_{AA} \end{pmatrix}$ to characterize an imaging system's detection efficiencies of donor and acceptor photons, the measured donor and acceptor intensity vector $\begin{pmatrix} d \\ a \end{pmatrix}$ is given by

$$\begin{pmatrix} d \\ a \end{pmatrix} = \begin{pmatrix} e_{DD} & e_{AA} \\ e_{AD} & e_{AA} \end{pmatrix} \begin{pmatrix} D \\ A \end{pmatrix} + \begin{pmatrix} c_D \\ c_A \end{pmatrix}, \qquad (5)$$

where $e_{DD}$: donor channel detection efficiency of donor emission,

$e_{DA}$: donor channel detection efficiency of acceptor emission (i.e., A-to-D bleed-through),

$e_{AD}$: acceptor channel detection efficiency of donor emission (i.e., D-to-A bleed-through),

$e_{AA}$: acceptor channel detection efficiency of acceptor emission,

$c_D$: camera count offset value in donor channel, and

$c_A$: camera count offset value in acceptor channel.

Equations above can be combined and rearranged as

$$\begin{pmatrix} d \\ a \end{pmatrix} = uq_D \begin{pmatrix} \bar{d}_{DD}(1-f) + \bar{d}_{AD}f \\ \bar{a}_{DD}(1-f) + \bar{a}_{AD}f \end{pmatrix} + uq_A \begin{pmatrix} \bar{d}_{AA} \\ \bar{a}_{AA} \end{pmatrix} + \begin{pmatrix} c_D \\ c_A \end{pmatrix}, \qquad (6)$$

where $\bar{d}_{DD} \equiv e_{DD}\bar{D}_0$: donor detection by donor channel in the absence of acceptor,

$\bar{d}_{AD} \equiv e_{DA}\bar{D}_0$: acceptor-to-donor channel bleed-through due to FRET when FRET = 1,

$\bar{a}_{DD} \equiv e_{AD}\bar{D}_0$: donor-to-acceptor channel bleed-through in the absence of acceptor,

$\bar{a}_{AD} \equiv e_{AA}\bar{D}_0$: acceptor detection by acceptor channel due to FRET when FRET = 1,

$\bar{d}_{AA} \equiv e_{DA}\bar{A}_0$: acceptor-to-donor channel bleed-through due to direct acceptor excitation,

$\bar{a}_{AA} \equiv e_{AA}\bar{A}_0$: acceptor detection by acceptor channel due to direct acceptor excitation,

$\bar{D}_0 \equiv \bar{Q}_D \bar{U}$: reference total donor emission in the absence of acceptor,

$\bar{A}_0 \equiv \bar{Q}_A \bar{U}$: reference total acceptor emission due to direct acceptor excitation.

Once the eight calibration parameters ($\bar{d}_{DD}$, $\bar{d}_{AD}$, $\bar{a}_{DD}$, $\bar{a}_{AD}$, $\bar{d}_{AA}$, $\bar{a}_{AA}$, $c_D$, $c_A$) are determined, Eq. (6) can be solved for ($q_D$, $f$) in terms of ($u$, $q_A$, $d$, $a$).

In our PIFE-FRET experiment, Cy3 and Atto647N were used as donor and acceptor dyes, and their fluorescence signals upon 561 nm laser excitation were simultaneously detected by splitting the camera field of view into a donor detection channel with 580−614 nm pass band and an acceptor detection channel with 656−734 nm pass band. The fluorescence bleed-through from acceptor-to-donor channel is almost completely prevented but the donor-to-acceptor channel bleed-through is not negligeable because of a tail towards longer wavelengths in the Cy3 emission spectrum. In this case, the condition $e_{DA} = 0$ leads to $\bar{d}_{AD} = \bar{d}_{AA} = 0$, reducing the number of calibration parameters from 8 to 6. The remaining 6 parameters could be determined by calibration experiments with 30-bp LSP DNA labeled with Atto647N and Cy3. Atto647N was labeled at −12-end but Cy3 was either internally labeled at −21 position or end labeled at −41-end of LSP DNA (Supplementary Fig. 5). Single-molecule donor/acceptor fluorescence traces from the two DNA constructs showed five types of events: both donor and acceptor bleaching, donor only bleaching, acceptor only bleaching, low FRET for end-labeled Cy3, and high FRET for mid-labeled Cy3 (Supplementary Fig. 5A−D). These events appeared as clusters in 2D plots of donor/acceptor signals constructed from many traces (Supplementary Fig. 5B, E) and the mean values of donor and acceptor fluorescence for each cluster was determined by 2D Gaussian mixture model-based clustering technique. To estimate the calibration parameters, 1) the normalized donor (or acceptor) quantum efficiency $q_D$ (or $q_A$) was assigned to 0 if the dye molecule was photobleached or to 1 if not photobleached; 2) the FRET value $f$ was assigned to $f_H$ and $f_L$ for the mid- and end-labeled Cy3 case, respectively; and 3) the normalized excitation power $u$ was assigned to 1 and $u_L$ for the mid- and end-labeled Cy3 case, respectively−although we maintained the excitation laser power and illumination angle the same through the experiments, small variations in the actual excitation power delivered to the fluorophores was inevitable when the mid- and end-labeled Cy3 samples were mounted on the microscope interchangeably one after another. The mean donor/acceptor fluorescence values for the five clustered events and the aforementioned choice of normalization conditions generated a system of coupled nonlinear equations that could be numerically solved to yield the 6 calibration parameters and the 3 additional unknowns, $f_H$, $f_L$, and $u_L$. The phenomenon of PIFE is unique to cyanine dyes like Cy3, and this corresponds to increase in the donor quantum efficiency $q_D$ upon binding of a protein nearby. In contrast, Atto647N does not exhibit PIFE effect and thus the acceptor quantum efficiency $q_A$ can be considered a constant. We therefore treated $q_D$, $f$, and $u$ as variables while fixing $q_A$ to 1 (if acceptor is not photobleached) and plugging in the estimated values for the calibration parameters into Eq. (6). Using this formalism, each single-molecule time trace of donor/acceptor fluorescence from the channel calibration data was converted to the time trace of $f$ and $q_D$ that could be termed 'Calibrated FRET' and 'PIFE'. The 2D histogram plots of $f$ vs $q_D$ from many traces showed $q_D \simeq 1$ regardless of FRET states (i.e., $f$ value), reconfirming a reliable calibration of the donor and acceptor detection channels (Supplementary Fig. 5C, F). Figure 2F, L in the main text were obtained similarly with extracting $f$ vs $q_D$ for LSP alone and LSP + TFAM cases from Eq. (6) while keeping all other parameters the same.

## Two conformational states model of DNA-TFAM interaction

Case 1. Single conformational state of the complex:

$$D + A \underset{k_{-1}}{\overset{k_1}{\rightleftharpoons}} DA, \qquad (7)$$

where $k_1$ and $k_{-1}$ are the association and dissociate rate constants of DNA ($D$) and TFAM ($A$) binding.

**Equilibrium binding affinity.** Consider DNA ($D$) of a constant concentration is titrated with TFAM ($A$) in equilibrium. The probability $P_{DA}$ of $D$ forming the complex $DA$ follows the well-known one-site ligand binding curve:

$$P_{DA} = \frac{[A]}{K_d + [A]}, \qquad (8)$$

where dissociation constant $K_d$ is defined by $K_d \equiv \frac{[D][A]}{[DA]} = \frac{k_{-1}}{k_1}$. For the known total concentrations $D_0$ and $A_0$ of DNA and TFAM, respectively, the concentration $[A]$ of the unbound TFAM in the expression of $P_{DA}$ is determined by solving the mass conservation equations:

$$D_0 = [D] + [DA] = [D] + \frac{1}{K_d}[D][A], A_0 = [A] + [DA] = [A] + \frac{1}{K_d}[D][A] \qquad (9)$$

**Dissociation rate.** Consider the stopped-flow experiment in which a very large amount of unlabeled DNA ($D$) is added to an equilibrium solution of TFAM ($A$) and labeled DNA ($D^*$). Then the labeled DNA that initially formed a complex with TFAM will dissociate over time according to the following reaction:

$$D^* + A \underset{k_{-1}}{\longleftarrow} D^*A. \qquad (10)$$

Solving the corresponding rate equation, $\frac{d}{dt}N_{D^*A} = -k_{-1}N_{D^*A}$, we obtain the decay curve of the number of $D^*A$ complex in time as

$$N_{D^*A}(t) \propto e^{-k_{-1}t}. \qquad (11)$$

Case 2. Two conformational states (i.e., P- and F-state) of the complex:

$$D + A \underset{k_{-1}}{\overset{k_1}{\rightleftharpoons}} (DA)_P \underset{k_{-2}}{\overset{k_2}{\rightleftharpoons}} (DA)_F, \qquad (12)$$

where $(DA)_P$ and $(DA)_F$ stand for the P- and F-state that transition between them with P-to-F and F-to-P rate constants $k_2$ and $k_{-2}$, respectively. The dissociation rate constant in this model corresponds to disassembly of the P-state and is denoted by $\tilde{k}_{-1}$ to differentiate it from Case 1.

**Equilibrium binding affinity.** Consider DNA ($D$) of a constant concentration is titrated with TFAM ($A$) in equilibrium. The probability $P_{DA}$ of $D$ forming the complex $DA$, accounting for both $(DA)_P$ and $(DA)_F$ states, can be derived using the dissociation constant $\tilde{K}_d \equiv \frac{[D][A]}{[(DA)_P]} = \frac{\tilde{k}_{-1}}{k_1}$ and the conformational equilibrium constant $K_2 \equiv \frac{[(DA)_F]}{[(DA)_P]} = \frac{k_2}{k_{-2}}$ as follows:

$$P_{DA} = P_{(DA)_P} + P_{(DA)_F} = \frac{[(DA)_P] + [(DA)_F]}{[D] + [(DA)_P] + [(DA)_F]} = \frac{[A]}{\tilde{K}_d/(1+K_2) + [A]}. \qquad (13)$$

This expression is formally the same as the one-site ligand binding curve in Case 1, if a composite (or apparent) dissociation constant $K_d$ is defined by $K_d \equiv \tilde{K}_d/(1+K_2)$.

**Dissociation rate.** The same stopped-flow experiment as in Case 1 leads to dissociation of the labeled DNA ($D^*$) that initially formed a complex with TFAM ($A$) according to the following kinetic scheme:

$$D^* + A \underset{\widetilde{k}_{-1}}{\leftarrow} \left(D^*A\right)_P \underset{k_{-2}}{\overset{k_2}{\rightleftharpoons}} \left(D^*A\right)_F. \tag{14}$$

Abbreviating the number of labeled DNA existing in the P-state and F-state at time $t$ to $N_P(t)$ and $N_F(t)$, respectively, they are determined by the following rate equations:

$$\frac{d}{dt}N_P(t) = -(\widetilde{k}_{-1} + k_2)N_P(t) + k_{-2}N_F(t)$$
$$\frac{d}{dt}N_F(t) = k_2 N_P(t) - k_{-2}N_F(t), \tag{15}$$

The solutions of $N_P(t)$ and $N_F(t)$ both follow double-exponential decay functions with the two decay rates $r_+$ and $r_-$ given by

$$r_\pm = \frac{\left|\widetilde{k}_{-1} + k_2 + k_{-2}\right| \pm \sqrt{\left(\widetilde{k}_{-1} + k_2 + k_{-2}\right)^2 - 4\widetilde{k}_{-1}k_{-2}}}{2}. \tag{16}$$

Here, $k_2 + k_{-2}$ corresponds to the equilibration rate between the P- and F-state (i.e., sum of bending and unbending rates) that is too fast ($\gtrsim \sim 25 \sec^{-1}$) to be captured by the ensemble fluorescence anisotropy-based stopped-flow assay. Under this condition, the fitted decay rate of an ensemble fluorescence anisotropy curve reflects the slower rate $r_-$ that can be further approximated by

$$r_- \approx \frac{\widetilde{k}_{-1}}{1 + K_2}, \tag{17}$$

when the dissociation rate $\widetilde{k}_{-1}$ is way slower than $k_2 + k_{-2}$.

### Reporting summary
Further information on research design is available in the Nature Portfolio Reporting Summary linked to this article.

### Data availability
Data supporting the findings of this study are available in the main text and the Supplementary Materials. Raw microscopy image data and donor-acceptor fluorescence time traces generated in this study have been deposited to Mendeley Data (https://doi.org/10.17632/4whngps32r.1). Source data are provided with this paper.

### Code availability
The custom-built software developed for analyzing and plotting smFRET and smPIFE data is deposited to Zenodo (https://doi.org/10.5281/zenodo.8378594). Supplementary Software 1 is also provided with this paper to help readers reproduce the relevant figures in this paper.

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

## Acknowledgements
We are grateful to Dr. Stephen Burley (Rutgers) for providing helpful feedback and insightful comments related to the project and the manuscript, and Dr. Craig Cameron (University of North Carolina at Chapel Hill) for the C-tail deleted TFAM expression plasmid. This work was funded by the National Institutes of Health Grant NIGMS MIRA grant GM118086 (to S.S.P.), Department of Energy grant DE-SC0019313 (to S.-H.L.), and Rutgers New Faculty Startup Fund through the Institute for Quantitative Biomedicine (to S.-H.L.).

## Author contributions
S.S.P. and S.-H.L. conceived the study. H.H. designed and conducted smFRET and smPIFE experiments. H.H. and S.-H.L. analyzed smFRET and smPIFE data. A.R. designed the smFRET substrates. J.S. and A.R. expressed and purified proteins. J.S., Y.A. and A.R. conducted ensemble fluorescence and biochemistry experiments. J.S., Y.A., S.S.P. and S.-H.L. analyzed ensemble experimental data. S.S.P. and S.-H.L. wrote the paper.

## Competing interests
The authors declare no competing interests.
