## [Peer Review File · Nature Communications]

Sequence-specific dynamic DNA bending explains mitochondrial TFAM's dual role in DNA packaging and transcription initiationReviewers' Comments:

Reviewer #1:

Remarks to the Author:

Patel and coworkers show that TFAM binding to mtDNA promoters, in particular to LSP, HSP and a non-specific sequence NS, results in two types of DNA bending: in a U-turn (full bending state, F) or in intermediate states (partial DNA bending, P). They find that in the P state the HMG-boxes are closer to the ends of the DNA. They also find that the binding to HSP results in lower FRET, thus lesser DNA bending, which is more pronounced for NS. They also see that the binding is more stable for LSP than for HSP or NS, as well as the dissociation rate from DNA by TFAM is much more reduced for LSP with respect the HSP and NS. They also discover the relationships between the association and dissociation rates between the off-state, the partially bend state and the full-bending state. The paper is technically sound, it uses and combines simultaneously multiple techniques that include FRET, stop-flow, and smPIFE, among others. The methods are explained in detail. The contribution is sound and clarifies an important question in the field, how TFAM combines its multiple DNA binding modes in mtDNA regulation of transcription and mtDNA packaging. The authors successfully address this question. The work is robust and supports the conclusions.

However, the text is highly technical and the reading is difficult for a broader audience that Nat Comm is intended for. Therefore the text of the results would greatly benefit from moving the technical terms to the mat & Meth, since it would facilitate the reading and help in a fluent connection between findings.

Introduction:

line 63: Several biophysical works substantiated the bending of the DNA by TFAM. However not all of them showed that nsDNA has a smaller bending angle, the sentence should be rephrased.

Results:

The first paragraph includes too many technicalities that should go to the Mat. & Meth, leaving only the ones indispensable for the understanding of the results. The same at the end of the second paragraph (line 123).

Paragraphs should have at least a one-sentence conclusion

L73-74: almost everybody in mol bio knows what is FRET, and much fewer people knows what is smPIFE. I would strongly recommend to explain well each technique in a single sentence and what is the aim in applying such a technique, to facilitate the broaden audience understanding. Also, regarding FRET, explain what is the basis for the correlation between "full bending" and high FRET, which is obvious for the specialists but not that much for non-specialists.

L115: please define dwell time analysis. I am not an English speaker: Is not more common to say "time of residence"?

Line 133: "The lack of severe bending may explain why the C-tail deletion mutant of TFAM 134 does not activate transcription". This sentence should include the results of the structure of TFAM bound to DNA, Factor B and RNAPol (Hillen Cell 2017)

Line 221: Please indicate from where the NS sequence is. This is important for the mtDNA audience and deserves to go to the main text. A radical friend of mine says that no DNA sequence is non-specific, all have features that condition protein binding. It is noticeable that the FRET values are closer to the ones of HSP.

Line 267: please define FITC and TAMRA at least with one or a couple of adjectives, as soon as they appear in the text but also when they recursively appear in different paragraphs.

L 283: "we introduced a TFAM and DNA interaction model that incorporated the two conformational states (Figure 4D upper panel)" – this model was introduced where?

L303: Please define in the text what it is ensemble fluorescence-anisotropy based assay, its basis and how is used.

L288: "the strongly bound F-state is protected from dissociation" – protected? Sound strange in this context. Who protects this F-state? Please rephrase

Discussion:

L457. Please consider: It is true that the crystal structure shows only binding to a short DNA. However, several studies structural and biophysical show that TFAM indeed uses two surfaces to generate dimers, the surface of HMG-A analysed by Ngo Nat Comm 2014, and the surface that includes part of the linker and the loop previous to the C-terminal end (and included in the so-called C-terminal tail), analysed in Uchida Elife 2017, formerly by Wong NAR 2009, and lately by Cuppari NAR 2019. This is a second important aspect regarding DNA binding, which is the induction of a protein conformation prone to dimerize. The authors should also comment on this.

L394: Please consider: ABF2 also induces a U-turn despite it is not involved in transcription activation (Chakraborty NAR 2017)

L473: Please consider: several experiments showed that RNA pol transcription initiation at non-specific sequences depends on the salt content of the solution. The group of Gustafsson has some data about this. In addition, proper RNAPol initiation needs the starting poly-Ade site at position +1. Finally, TFAM recursively generates U-turns when a minimal motif such as Gua-10bp-Gua is found (Choi NAR 2022).

Minor changes:

Line 111: 'Previous electron microscopy study had revealed...' is missing an 'A' such as: 'A previous...' – we suggest a revision of the English

Reviewer #2:

Remarks to the Author:

Mitochondrial DNA (mtDNA) consists of two promoter sites for transcription: light strand promoter (LSP) and heavy strand promoter (HSP). Mitochondrial transcription factor A (TFAM) is responsible for packaging mtDNA and also initiating transcription. One of the outstanding questions in the field is how TFAM differentiates between promoter and non-promoter sequences for transcription initiation, while conducting DNA packaging in a sequence non-specific manner. The authors used single molecule and bulk fluorescence assays to study the real-time dynamics of DNA bending and unbending upon TFAM binding. Their data suggests that the stability of the bent state of the sequence bound to TFAM determines the affinity and kinetic stability of the binding. The authors hypothesize that TFAM initially induces a partially bent DNA conformation, that can transition to a fully bent state depending on the DNA sequence. The fully bent state results in tighter binding and stability of the TFAM-DNA complex, inducing transcription initiation. LSP was shown to have 14-fold more stability in the fully bound state compared to HSP and NS, and the NS has a long residence time in the partially bent state.

This study builds upon previous structural and biochemical findings which demonstrated the bent conformation of DNA induced by TFAM. The merit of the current study is the demonstration of the kinetic, therefore the thermodynamic differences between different TFAM-promoter sequences. The study is limited in several ways (i) only a small subset of sequences was tested as promoter, non-

promoter sequence. (ii) the final model about the TFAM binding for mtDNA packaging vs. transcription is not supported at all by the results of this study. (iii) related to (ii), the kinetic difference between LSP, HSP and NS does not explain how the different modes of binding (P vs. F) contribute to the biological pathways.

Some questions and comments include:

1. What about the LSP sequence vs HSP and NS induces a more stable fully bent state? What causes the differences in inherent bendability among the sequences?
2. What is the effect of salt concentration on the ability of TFAM to bend the different DNA sequences?
3. Were other NS tested to determine the nature of binding with TFAM? Since the stability and kinetics of the TFAM-bound state is sequence dependent, are there possible NS that could be more bendable than others and result in greater TFAM binding affinity?
4. A new promoter, LSP2, was recently discovered in human mtDNA (DOI:<https://doi.org/10.1016/j.molcel.2022.08.011>). Have the dynamics of the TFAM-LSP2 complex been studied? Are they consistent with the model outlined in the paper?

Reviewer #3:

Remarks to the Author:

In this manuscript, Huh et al. use single-molecule as well as ensemble FRET experiments in conjunction with other biophysical techniques to characterize the binding and bending of DNA by the mitochondrial protein TFAM. Using these methods, they aim to address the fundamental conundrum of how this protein can on the one hand act as a general compaction factor for mtDNA and at the same time as a specific transcriptional activator. They propose a novel intermediate DNA bending mode of TFAM (P-state) and show that TFAM adopts the fully bent F-state more readily on the strong LSP promoter than on non-specific sequences. Based on their experiments, they propose a model of how TFAM can achieve both mtDNA compaction and POLRMT recruitment depending on the bound sequence.

This manuscript is very well written and the data is presented in a clear and concise fashion. The authors further discuss their data very extensively in the context of available structural and biophysical data. I am not an expert in the methods used in this manuscript, and I therefore cannot assess the technical aspects of experimental design and data analysis. However, I believe this manuscript provides exciting new insights into the mechanism how TFAM interacts with mtDNA, which warrant publication. I only have some comments to the authors:

- One seemingly contradictory aspect (which the authors also discuss) in the proposed model is that binding and bending at HSP is much weaker than at LSP, and only differs quite marginally from non-specific binding. This raises the question how efficient transcription can be achieved at HSP. This is especially interesting since the HSP-transcript carries most genes in mtDNA. There is a large body of in vitro transcription data published. Could the authors comment whether the observed differences in their experiments correlate with the strengths of the two promoters in in vitro assays?
- Related to the previous point, there was recently a third promoter proposed in human mtDNA (LSP2; Tan et al., Mol Cell 2022). Have the authors investigated the bending behaviour at this promoter? While I think it is not absolutely critical for this manuscript to do this experiment, I think it could substantiate the proposed model and provide further insights into how bending of DNA relates to transcriptional activity.

- One aspect that I was missing in the discussion of the potential relation between the observed bending behaviour and transcriptional activity was the role of POLRMT and TFB2M. Is the authors rightly state, interactions between POLRMT and the promoter DNA are involved in initiation complex formation. Could it be that these interactions contribute substantially to the transcriptional activity, especially at HSP where the F state is less stable? What effect does addition of POLRMT and TFB2M have on the kinetics and stability of P and F state formation?

AUTHOR RESPONSES TO REVIEWER COMMENTS

Note: To clearly indicate the line numbers where texts are modified, we have indicated both the line numbers corresponding to the original and the revised manuscripts with 'OL' and 'RL' prefixes, respectively: OL123-125 and RL132-138 for example. Those lines of changes in the revised manuscript are also highlighted yellow for better identification.

Reviewer #1 (Remarks to the Author):

General Comment:

Patel and coworkers show that TFAM binding to mtDNA promoters, in particular to LSP, HSP and a non-specific sequence NS, results in two types of DNA bending: in a U-turn (full bending state, F) or in intermediate states (partial DNA bending, P). They find that in the P state the HMG-boxes are closer to the ends of the DNA. They also find that the binding to HSP results in lower FRET, thus lesser DNA bending, which is more pronounced for NS. They also see that the binding is more stable for LSP than for HSP or NS, as well as the dissociation rate from DNA by TFAM is much more reduced for LSP with respect the HSP and NS. They also discover the relationships between the association and dissociation rates between the off-state, the partially bend state and the full-bending state. The paper is technically sound, it uses and combines simultaneously multiple techniques that include FRET, stop-flow, and smPIFE, among others. The methods are explained in detail. The contribution is sound and clarifies an important question in the field, how TFAM combines its multiple DNA binding modes in mtDNA regulation of transcription and mtDNA packaging. The authors successfully address this question. The work is robust and supports the conclusions.

However, the text is highly technical and the reading is difficult for a broader audience that Nat Comm is intended for. Therefore the text of the results would greatly benefit from moving the technical terms to the mat & Meth, since it would facilitate the reading and help in a fluent connection between findings.

General Response: Thank you for carefully reading our manuscript and providing constructive ideas for improving it. We have made all the requested changes as detailed below.

Comment 1-1: Introduction:

line 63: Several biophysical works substantiated the bending of the DNA by TFAM. However not all of them showed that nsDNA has a smaller bending angle, the sentence should be rephrased.

Response: We have modified the sentences:

Original (OL62-64)	TFAM-induced DNA bending in solution was demonstrated by ensemble biochemical and biophysical characterization ^{6,11,13,16} , which indicated the average bend angle was smaller for NS than LSP or HSP.
Revised (RL62-64)	TFAM-induced DNA bending in solution was demonstrated by ensemble biochemical and biophysical characterization ^{6,11,13,16} , and some of these studies indicated the average bend angle was smaller for NS than LSP or HSP.

Comment 1-2: Results:

The first paragraph includes too many technicalities that should go to the Mat. & Meth, leaving only the ones indispensable for the understanding of the results. The same at the end of the second paragraph (line 123).

Response: We have modified the sentences according to the comment. The technical details have been moved to the Figure 1 and S2 legends.

Original (OL92-95)	A 30-bp DNA fragment with the LSP sequence from -12 to -41 was chemically synthesized (referred to as 'LSP') (Figure 1A). To monitor DNA conformational dynamics with smFRET, LSP was end-labeled with Atto565 (donor) and Atto647N (acceptor) at -41 and -12 positions, respectively, and with (PEG-18)2-biotin at the -41 end for surface immobilization (Figure 1B).
Revised (RL97-99)	To monitor DNA conformational dynamics with smFRET, a 30bp LSP sequence was end-labeled with Atto565 (donor) and Atto647N (acceptor) at -41 and -12 positions, respectively, and with (PEG-18)2-biotin at the -41 end for surface immobilization (Figure 1A, 1B).
Original (OL115-123)	We quantified the two-state kinetics through dwell time analysis (Figure 1F). The F-state dwell time distribution fits poorly in a single-exponential model but fits well in a double-exponential model with a mean dwell time of 1.0 (\pm 0.3) sec (Figure S2). Although the P-state dwell time distribution fits a single-exponential model fairly well, a double-exponential model with a mean dwell time of 0.04 (\pm 0.01) sec nevertheless improves the fitting, especially in the long tail region of the distribution (Figure S2). The two-state kinetic model of DNA bending in the LSP-TFAM complex is graphically summarized in Figure 1I, where the forward rate constant (\sim 24 s ⁻¹) and the reverse rate constant (\sim 1 s ⁻¹) were obtained by taking the inverse of P-state mean dwell time and F-state mean dwell time, respectively.
Revised (RL119-124)	We quantified the two-state kinetics through fitting double-exponential models to the distribution of residence time (commonly termed 'dwell time') in each state (Figure 1F, S2), and obtained 1.0 (\pm 0.3) sec and 0.04 (\pm 0.01) sec as the average dwell times for the F- and P-state, respectively, showing a substantially higher stability in the F-state. The corresponding two-state kinetic model, summarized in Figure 1I, shows a much faster forward rate (\sim 24 s ⁻¹) than the reverse rate (\sim 1 s ⁻¹).

Comment 1-3: L73-74: *almost everybody in mol bio knows what is FRET, and much fewer people knows what is smPIFE. I would strongly recommend to explain well each technique in a single sentence and what is the aim in applying such a technique, to facilitate the broaden audience understanding. Also, regarding FRET, explain what is the basis for the correlation between "full bending" and high FRET, which is obvious for the specialists but not that much for non-specialists.*

Response: We have modified the sentences by adding sentences explaining smFRET and smPIFE techniques in the last paragraph of Introduction.

Original (OL72-75)	In this work, we studied the real-time dynamics of TFAM-induced DNA bending and unbending conformational changes at the single-molecule level using FRET (smFRET) and single-molecule protein-induced fluorescence enhancement (smPIFE) assays, in combination with ensemble measurements.
--

Revised (RL72-81)	In this work, we studied the real-time dynamics of TFAM-induced DNA bending and unbending conformational changes at the single-molecule level using smFRET and single-molecule protein-induced fluorescence enhancement (smPIFE) assays, in combination with ensemble measurements. In smFRET assay, ratiometric measurement of fluorescence from donor and acceptor that are attached to the ends of a DNA reports the real-time change in the end-to-end distance accompanied by DNA bending: the larger the bend angle, the shorter the end-to-end distance, resulting in brighter acceptor fluorescence and thus higher FRET value. In contrast, smPIFE allows to monitor the proximity of a protein to a specific position on DNA where a PIFE-compatible dye (e.g., cyanine dye Cy3) is attached and reports the protein-DNA distance by the fluorescence intensity.
-------------------	--

Comment 1-4: L115: please define dwell time analysis. I am not an English speaker: Is not more common to say “time of residence”?

Response: ‘Dwell time’ is a well-established terminology in single-molecule research communities. We appreciate the reviewer’s suggestion and have further clarified its meaning for a broader audience:

Original (OL115)	We quantified the two-state kinetics through dwell time analysis (Figure 1F).
Revised (RL119-120)	We quantified the two-state kinetics by fitting double-exponential models to the distribution of residence time (commonly termed ‘dwell time’) in each state (Figure 1F, S2), ...

Comment 1-5: Line 133: “The lack of severe bending may explain why the C-tail deletion mutant of TFAM does not activate transcription”. This sentence should include the results of the structure of TFAM bound to DNA, Factor B and RNAPol (Hillen Cell 2017)

Response: We have included the results of Hillen Cell 2017 by adding a sentence to Line 133 as follows:

Original (OL133)	... to the F-state. The lack of severe bending may explain why the C-tail ...
Revised (RL134-137)	... to the F-state. Structural studies of TFAM complexed with POLRMT and TFB2M ¹⁰ suggest that TFAM must be fully bent to interact with POLRMT. These interactions enable TFAM to recruit POLRMT and accurately position it over the transcription start site for de novo RNA synthesis. The lack of severe bending may explain why the C-tail ...

Comment 1-6: Line 221: Please indicate from where the NS sequence is. This is important for the mtDNA audience and deserves to go to the main text. A radical friend of mine says that no DNA sequence is non-specific, all have features that condition protein binding. It is noticeable that the FRET values are closer to the ones of HSP.

Response: The NS sequence is adopted from a previously published paper (Malarkey et al., 2012, DOI:10.1093/nar/gkr787) and originates from Mus musculus mitochondrion (NCBI

Sequence ID: NC_005089.1, from coordination 5684 to 5708) as shown below.

Mus musculus mitochondrion, complete genome

Sequence ID: NC_005089.1 Length: 16299 Number of Matches: 1

Range 1: 5684 to 5708 GenBank Graphics

▼ Next Match ▲ Pre

Score	Expect	Identities	Gaps	Strand
50.1 bits(25)	2e-05	25/25(100%)	0/25(0%)	Plus/Plus
Query 1	AGCAGGAGCAGGAACAGGATGAACA	25		
Sbjct 5684	AGCAGGAGCAGGAACAGGATGAACA	5708		

We have provided this information to Methods section as follows:

Original (OL639)	All DNA oligonucleotides were custom synthesized and purified by ...
Revised (RL681-683)	The NS sequence is adopted from a previously published paper ¹³ and originates from Mus musculus mitochondrion (NCBI Sequence ID: NC_005089.1, from coordination 5684 to 5708). All DNA oligonucleotides were custom synthesized and purified by ...

Comment 1-7: please define FITC and TAMRA at least with one or a couple of adjectives, as soon as they appear in the text but also when they recursively appear in different paragraphs.

Response: We have provided the full names 'Fluorescein isothiocyanate' and 'Tetramethylrhodamine' for FITC and TAMRA, respectively, in OL267 (RL271) where the acronyms appear for the first time.

Comment 1-8: L 283: "we introduced a TFAM and DNA interaction model that incorporated the two conformational states (Figure 4D upper panel)" – this model was introduced where?

Response: We have modified the sentence.

Original (OL282-284)	To ascribe the slow-off rate of LSP-TFAM to the stability of the F-state, we introduced a TFAM and DNA interaction model that incorporated the two conformational states (Figure 4D upper panel).
Revised (RL286-289)	The slow off rate can be explained by a two-step DNA-TFAM interaction model that involves the DNA bending conformational change after the bimolecular TFAM and DNA binding step (Figure 4D, upper panel)

Comment 1-9: L303: Please define in the text what it is ensemble fluorescence-anisotropy based assay, its basis and how is used.

Response: We have modified the sentences.

Original (OL302-305)	We next studied the sequence-dependence of equilibrium binding affinity between TFAM and DNA using an ensemble fluorescence-anisotropy based assay. The DNA-TFAM complex is much larger than DNA; hence, if we label DNA with FITC, we can measure DNA-protein interactions through the increased anisotropy of labeled DNA upon complex formation (Figure 4E).
Revised (RL306-313)	We next studied the sequence-dependence of equilibrium binding affinity between TFAM and DNA using an ensemble fluorescence anisotropy-based assay. The rotational mobility of the fluorescent molecule influences the fluorescence anisotropy value. An increase in the size of the molecule to which a fluorophore is attached leads to reduced rotational mobility, resulting in higher fluorescence anisotropy. Conversely, smaller molecules exhibit more free rotation, leading to a lower anisotropy value. The DNA-TFAM complex is much larger than DNA; hence, if we label DNA with FITC, we can measure DNA-protein interactions through the increased anisotropy of labeled DNA upon complex formation (Figure 4E).

Comment 1-10: *“the strongly bound F-state is protected from dissociation” – protected? Sound strange in this context. Who protects this F-state? Please rephrase*

Response: We have modified the sentences.

Original (OL288-289)	... the strongly bound F-state is protected from dissociation but can transform back to the P-state with an “unbending rate” ($k_{F \rightarrow P}$).
Revised (RL292-293)	... the F-state can transform back to the P-state without dissociation with an “unbending rate” ($k_{F \rightarrow P}$).

Comment 1-11: *Discussion: L457. Please consider: It is true that the crystal structure shows only binding to a short DNA. However, several studies structural and biophysical show that TFAM indeed uses two surfaces to generate dimers, the surface of HMG-A analysed by Ngo Nat Comm 2014, and the surface that includes part of the linker and the loop previous to the C-terminal end (and included in the so-called C-terminal tail), analysed in Uchida Elife 2017, formerly by Wong NAR 2009, and lately by Cuppari NAR 2019. This is a second important aspect regarding DNA binding, which is the induction of a protein conformation prone to dimerize. The authors should also comment on this.*

Response: We have modified the following sentences to incorporate the reviewer’s comment.

Original (OL460-461)	Such studies will be important in better understanding how TFAM interacts with the mtDNA in physiological conditions.
Revised (RL480-483)	Several studies show that DNA-bound TFAM can dimerize using the surface of HMG-A ¹⁶ or the C-tail ^{14,22,39} . Therefore, the extension of our single-molecule study to longer DNA will be important in better understanding how TFAM interacts with the mtDNA in physiological conditions.

Comment 12: L394: Please consider: ABF2 also induces a U-turn despite it is not involved in transcription activation (Chakraborty NAR 2017)

Response: We have modified the following sentences to incorporate the reviewer's comment.

Original (OL394-396)	Interestingly, ABF2, the yeast homolog of TFAM, involved in mtDNA packaging but not for transcription activation, lacks the C-tail and shows a shallow average DNA bend of 78° corresponding more closely to the P-state of DNA-TFAM ³² .
Revised (RL408-413)	Interestingly, ABF2, the yeast homolog of TFAM, involved in mtDNA packaging but not for transcription activation, lacks the C-tail, which likely affects the stability of its fully bent state. A published structure of ABF2 ³² shows that each HMG-box induces a 90° bend in the DNA. However, since this depiction is from a crystal structure, it might exclusively capture the stable fully bent state. In contrast, an AFM study ³³ measured an average DNA bending angle of 78° from 43 DNA-TFAM complexes, encompassing various DNA bending angles.

Comment 1-13: Please consider: several experiments showed that RNA pol transcription initiation at non-specific sequences depends on the salt content of the solution. The group of Gustafsson has some data about this. In addition, proper RNAPol initiation needs the starting poly-Ade site at position +1. Finally, TFAM recursively generates U-turns when a minimal motif such as Gua-10bp-Gua is found (Choi NAR 2022).

Response: Gustafsson's experiments (Posse et al.,2017, DOI: DOI: 10.1074/jbc.M116.751008) primarily employed mouse POLRMT, not human POLRMT. In the context of human POLRMT, a study by (Uchida et al.,2017,DOI: 10.7554/eLife.27283) revealed that randomizing the TFAM-binding site in either LSP or HSP resulted in a substantial decrease in transcription activity, highlighting the essential role of sequence specificity for transcription.

We have modified the following sentences to incorporate the reviewer's comment.

Original (OL472-475)	Last but not least, ... between HSP-TFAM and NS-TFAM complexes.
Revised (RL494-503)	Last but not least, ... between HSP-TFAM and NS-TFAM complexes. A study (Uchida 2017) showed that randomizing the TFAM-DNA binding sequence in either LSP or HSP substantially decreased transcription activity, highlighting the essential role of the upstream promoter sequence for transcription initiation. A recent study (Choi 2022) pointed out that TFAM interacts with the guanines in the GN10G motif in the TFAM DNA binding site, and this guanine pair is placed at an identical distance (-20/-31) from the transcription start sites in LSP and HSP. This suggests that TFAM must bend the DNA at a precise location upstream of the start site for optimal transcription initiation by POLRMT, which is not the case for NS.

Comment 1-14: Minor changes:

Line 111: 'Previous electron microscopy study had revealed... ' is missing an 'A' such as: 'A previous...' – we suggest a revision of the English

Response: We have modified the sentence accordingly.

Reviewer #2 (Remarks to the Author):

General Comment:

Mitochondrial DNA (mtDNA) consists of two promoter sites for transcription: light strand promoter (LSP) and heavy strand promoter (HSP). Mitochondrial transcription factor A (TFAM) is responsible for packaging mtDNA and also initiating transcription. One of the outstanding questions in the field is how TFAM differentiates between promoter and non-promoter sequences for transcription initiation, while conducting DNA packaging in a sequence non-specific manner. The authors used single molecule and bulk fluorescence assays to study the real-time dynamics of DNA bending and unbending upon TFAM binding. Their data suggests that the stability of the bent state of the sequence bound to TFAM determines the affinity and kinetic stability of the binding. The authors hypothesize that TFAM initially induces a partially bent DNA conformation, that can transition to a fully bent state depending on the DNA sequence. The fully bent state results in tighter binding and stability of the TFAM-DNA complex, inducing transcription initiation. LSP was shown to have 14-fold more stability in the fully bound state compared to HSP and NS, and the NS has a long residence time in the partially bent state.

This study builds upon previous structural and biochemical findings which demonstrated the bent conformation of DNA induced by TFAM. The merit of the current study is the demonstration of the kinetic, therefore the thermodynamic differences between different TFAM-promoter sequences. The study is limited in several ways (i) only a small subset of sequences was tested as promoter, non-promoter sequence. (ii) the final model about the TFAM binding for mtDNA packaging vs. transcription is not supported at all by the results of this study. (iii) related to (ii), the kinetic difference between LSP, HSP and NS does not explain how the different modes of binding (P vs. F) contribute to the biological pathways.

General Response: We thank the reviewer for the critical evaluation of our work. As pointed out, this is the first study that establishes new single-molecule methods to study TFAM-DNA bending dynamics. Using these methods, we are able to show differences between LSP, HSP, and a NS DNA sequence that were not known previously. This knowledge will provide a good foundation to address the gaps that the reviewer has listed while incorporating newer studies such as the minimal motif (Choi and Garcia-Diaz 2022) and LSP2 sequence (Tan et al 2022). We also acknowledge the limitations listed by the reviewer. Note that our discussion indeed points out many of the limitations in terms of other NS sequences, our use of short DNAs, the need to develop methods to study TFAM bending of long DNAs, and the role of TFAM-DNA bending in transcription initiation. We have further elucidated both the novelty and the limitations of our work through responses to the specific questions and issues raised by the reviewer as follow.

Comment 2-1: *Some questions and comments include:*

1. What about the LSP sequence vs HSP and NS induces a more stable fully bent state? What causes the differences in inherent bendability among the sequences?

Response: This is an important question but difficult to fully answer as we currently lack sufficient information. However, the basic sequence comparison of LSP, HSP, and NS provides clues about why LSP induces a more stable, fully bent state than HSP and NS sequences.

1. A recent study [1] pointed out that a GN₁₀G sequence motif is required for high-affinity TFAM binding. This motif is present in LSP, HSP, and the NS DNA that was used in our studies (see

figure below). Our analysis indicates the pair of guanines in the GN₁₀G motif of LSP contains 5'-TG-3'/5'-CA-3' dinucleotide steps (**in red**), which are more deformable base steps according to literature [2-4]. Previously published data pointed out that TG/CA dinucleotide steps act as a better target site for high mobility group proteins [5].

2. Further, we also found that adjacent to these 5'-TG-3'/5'-CA-3' steps, there are 5'-TT-3'/5'-AA-3' steps present in the LSP (**green box**). This AT-rich sequence provides higher flexibility (due to low base stacking energy) and creates a perfect kink that can stabilize the bend conformation [6]. Unlike LSP, the HSP lacks one of the bona fide 5'-TG-3'/5'-CA-3' steps at the downstream end and lacks the TT/AA dinucleotide step around the TG/CA step. The NS sequence contains a GN₁₀G motif (**scheme 1**) but lacks the bona fide TG/CA motif and the TT/AA dinucleotide step (**scheme 2**). Published structural studies show additional base contacts in the TFAM-LSP complex that are absent in HSP-TFAM and NS-TFAM complexes [7].
3. Additionally, LSP has an A...T rich downstream end sequence compared with the HSP and NS sequence (**dashed brown box**), which may stabilize the fully bent state. Previous studies stated that the CA/TG dinucleotide step in the 5' end of the oligo A tract strongly modulates the intrinsic DNA bendability [8].
4. Finally, the stacking energy of each dinucleotide step in LSP, HSP, and NS sequences gives us a clue about the overall flexibility of these sequence [6]. The LSP has the lowest average stacking energy of all the dinucleotide steps (-7.27 kcal/mol), followed by NS (-8.17 kcal/mol) and HSP (-8.61 kcal/mol).

This analysis creates testable hypotheses. However, testing these hypotheses will require a systematic examination of a series of DNA sequences (mutating the TG/CA and TT/AA dinucleotide steps) coupled with smFRET studies to measure the precise DNA bending dynamics. We believe that such studies are currently out of scope for this manuscript.

We have incorporated these ideas in Supplementary Information by adding Supplementary Texts, Figure S7, and Supplementary References. We have also modified sentences in Discussion to provide a brief summary of the ideas as follow (see the boldfaced sentences).

Original (OL344)	... affinity of TFAM for DNA (Figure 4C and 4G). Based on these findings, we propose a model ...
Revised (RL352-360)	... affinity of TFAM for DNA (Figure 4C and 4G). Our basic sequence analysis shows that the coexistent high-affinity TFAM-binding sequence motif and multiple sites of higher intrinsic DNA bendability are found only in LSP (see Supplementary Texts, Figure S7), which may provide clues to the sequence-dependent stability of the F-state. Interestingly, lowering salt (NaCl) concentration interrupts the F-state stability of LSP-TFAM complex (Figure S8), indicating salt-dependence of the aforementioned site-specific TFAM interactions with LSP and hence possible regulatory mechanisms of mitochondrial transcription. Based on these findings, we propose a model ...

Supplementary References

1. Choi, W.S. and M. Garcia-Diaz, *A minimal motif for sequence recognition by mitochondrial transcription factor A (TFAM)*. *Nucleic Acids Research*, 2022. **50**(1): p. 322-332.
2. Travers, A.A., *DNA bending and kinking—sequence dependence and function: Current Opinion in Structural Biology 1991*, 1: 114–122. *Current Opinion in Structural Biology*, 1991. **1**(1): p. 114-122.

3. Olson, W.K. and V.B. Zhurkin, *Working the kinks out of nucleosomal DNA*. Current opinion in structural biology, 2011. **21**(3): p. 348-357.
4. Ojha, R.P., et al., *DNA bending and sequence-dependent backbone conformation: NMR and computer experiments*. European journal of biochemistry, 1999. **265**(1): p. 35-53.
5. Churchill, M., et al., *HMG-D is an architecture-specific protein that preferentially binds to DNA containing the dinucleotide TG*. The EMBO Journal, 1995. **14**(6): p. 1264-1275.
6. Ussery, D.W., *DNA Structure: A-, B-and Z-DNA Helix Families*. Encyclopedia of life sciences, 2002. **1**: p. e003122.
7. Ngo, H.B., et al., *Distinct structural features of TFAM drive mitochondrial DNA packaging versus transcriptional activation*. Nature communications, 2014. **5**(1): p. 3077.
8. Nagaich, A.K., et al., *CA/TG sequence at the 5' end of oligo (A)-tracts strongly modulates DNA curvature*. Journal of biological chemistry, 1994. **269**(10): p. 7824-7833.

Comment 2-2: 2. What is the effect of salt concentration on the ability of TFAM to bend the different DNA sequences?

Response: To address this important issue, we have analyzed a smFRET dataset that had been previously acquired for LSP and NS at two different NaCl concentrations, 50 mM and 2.5 mM. The results are included in a supplementary figure (Figure S8) that is shown below for convenience. We find that whereas LSP-TFAM is highly stable in the fully bent F-state at 50 mM NaCl, lowering NaCl concentration to 2.5 mM introduces time intervals when the F-state gets

unstable while the P-state gets more stable. This results in substantial increase of the mean P-state dwell time from 0.02 sec to 0.4 sec, while decreasing the mean F-state dwell time from 1.4 sec to 0.9 sec in this dataset. In contrast, NS appears insensitive to NaCl concentration. This indicates that the site-specific TFAM interactions with LSP, which are mentioned above in the response to the reviewer's previous comment (*Comment 2-1*), may be salt dependent.

Effects of NaCl concentration on DNA-TFAM bending dynamics. This figure is included in Figure S8.

Additionally, these new data and the results are referenced at the end of the first paragraph of Discussion section by adding a sentence as follows (see the boldfaced sentences).

Original (OL344)	... affinity of TFAM for DNA (Figure 4C and 4G). Based on these findings, we propose a model ...
Revised (RL352-360)	... affinity of TFAM for DNA (Figure 4C and 4G). Our basic sequence analysis shows that the coexistent high-affinity TFAM-binding sequence motif and multiple sites of higher intrinsic DNA bendability are found only in LSP (see Supplementary Texts, Figure S7), which may provide clues to the sequence-dependent stability of the F-state. Interestingly, lowering salt (NaCl) concentration interrupts the F-state stability of LSP-TFAM complex (Figure S8), indicating salt-dependence of the aforementioned site-specific TFAM interactions with LSP and hence possible regulatory mechanisms of mitochondrial transcription.
	Based on these findings, we propose a model ...

Comment 2-3: 3. Were other NS tested to determine the nature of binding with TFAM? Since the stability and kinetics of the TFAM-bound state is sequence dependent, are there possible NS that could be more bendable than others and result in greater TFAM binding affinity?

Response: The NS sequence in this study was adapted from a previously published paper by the Churchill group, which conducted a systematic comparison of LSP, HSP, and NS bending by ensemble FRET (Malarkey et al., 2012, DOI:10.1093/nar/gkr787). Not pointed out by the original paper, we found that this sequence originates from the *Mus musculus* mitochondrial DNA (NCBI Sequence ID: NC_005089.1, from coordination 5684 to 5708).

As pointed out above, this NS happens to contain a GN10G motif that was recently pointed out by Choi and Garcia-Diaz (NAR 2022) as a minimal motif for high-affinity TFAM binding. We were not aware of the Choi study when choosing our NS DNA, and to answer the reviewer's question, we have not examined other NS sequences. Given Choi's study, it will indeed be interesting to examine DNA sequences that lack this minimal motif using single-molecule DNA bending studies and compare the dynamics of bending to HSP. We hope to carry out such studies in the future.

Comment 2-4: 4. A new promoter, LSP2, was recently discovered in human mtDNA (DOI:<https://doi.org/10.1016/j.molcel.2022.08.011>). Have the dynamics of the TFAM-LSP2 complex been studied? Are they consistent with the model outlined in the paper?

Response: We are excited to measure the DNA bending dynamics of the LSP2 sequence, reported in this recent publication, and compare it to LSP and HSP. However, this study came out after the completion of our study, and I am currently on sabbatical (Sang-Hyuk Lee), and Hyun Huh, who conducted the single molecule studies, has moved on to another position. Although ensemble studies can be carried out with LSP2, we believe that the differences between LSP1 and LSP2 will be apparent only in single-molecule DNA dynamic studies. We expect LSP2 to behave like HSP, given its poor transcription activity compared to LSP, as reported by Tan et al. (2022).

Reviewer #3 (Remarks to the Author):

General Comment: *In this manuscript, Huh et al. use single-molecule as well as ensemble FRET experiments in conjunction with other biophysical techniques to characterize the binding and bending of DNA by the mitochondrial protein TFAM. Using these methods, they aim to address the fundamental conundrum of how this protein can on the one hand act as a general compaction factor for mtDNA and at the same time as a specific transcriptional activator. They propose a novel intermediate DNA bending mode of TFAM (P-state) and show that TFAM adopts the fully bent F-state more readily on the strong LSP promoter than on non-specific sequences. Based on their experiments, they propose a model of how TFAM can achieve both mtDNA compaction and POLRMT recruitment depending on the bound sequence.*

This manuscript is very well written and the data is presented in a clear and concise fashion. The authors further discuss their data very extensively in the context of available structural and biophysical data. I am not an expert in the methods used in this manuscript, and I therefore cannot assess the technical aspects of experimental design and data analysis. However, I believe this manuscript provides exciting new insights into the mechanism how TFAM interacts with mtDNA, which warrant publication. I only have some comments to the authors:

General Response: Thank you for assessing our manuscript positively and providing thoughtful comments. We have provided our response to each comment below.

Comment 3-1:- *One seemingly contradictory aspect (which the authors also discuss) in the proposed model is that binding and bending at HSP is much weaker than at LSP, and only differs quite marginally from non-specific binding. This raises the question how efficient transcription can be achieved at HSP. This is especially interesting since the HSP-transcript carries most genes in mtDNA. There is a large body of in vitro transcription data published. Could the authors comment whether the observed differences in their experiments correlate with the strengths of the two promoters in in vitro assays?*

Response: There is a correlation between the stable bending of LSP and higher transcription efficiency from the following studies:

1. Uchida et al., (2017, DOI: 10.7554/eLife.27283) used the dual LSP-HSP promoter DNA fragment and demonstrated, through in vitro transcription assays, that under optimal conditions, the strength of transcription from LSP is higher than that of HSP: transcription from LSP reached ~500 nM, HSP1 reached a maximum value of ~300 nM (see the figure attached below).
2. In recent studies (Yan, et al., 2022, DOI: 0.1101/gr.275784.121) and (Tan et al., 2022, DOI: 10.1016/j.molcel.2022.08.011), utilizing in vivo ReCappable-seq and capable-seq analyses, respectively, revealed a significantly higher abundance of transcripts from LSP compared to HSP. One study reported ~10-fold higher amounts of transcription initiation transcripts from LSP than HSP and the other study reported a ~4-fold higher.

This information along with the three references are added to the discussion as follows.

Original (OL448-452)	Specific transcription on HSP and LSP is also achieved by sequence-specific interactions of POLRMT with the promoter initiation region ³⁸ . Future work needs to determine...
Revised (RL465-472)	Specific transcription on HSP and LSP is also achieved by sequence-specific interactions of POLRMT with the promoter initiation region ³⁹ . Recent studies ^{14,39,40} showed higher in vitro transcription efficiency for LSP than HSP, reporting ~4- to ~10-fold higher amounts of transcription initiation transcripts from LSP than HSP; hence, these results are nevertheless consistent with the correlation between the stable bending of LSP and its higher transcription efficiency as proposed in our model. Future work needs to determine ...

Comment 3-2:- Related to the previous point, there was recently a third promoter proposed in human mtDNA (LSP2; Tan et al., Mol Cell 2022). Have the authors investigated the bending behaviour at this promoter? While I think it is not absolutely critical for this manuscript to do this experiment, I think it could substantiate the proposed model and provide further insights into how bending of DNA relates to transcriptional activity.

Response: A similar comment was also made by reviewer 2 (*Comment 2-4*). We repeat our response here:

We are excited to measure the DNA bending dynamics of the LSP2 sequence, reported in this recent publication, and compare it to LSP and HSP. However, this study came out after the completion of our study, and I am currently on sabbatical (Sanghyuk Lee), and Hyun Huh, who conducted the single molecule studies, has moved on to another position. Although ensemble studies can be carried out with LSP2, we believe that the differences between LSP1 and LSP2

will be apparent only in single-molecule DNA dynamic studies. We expect LSP2 to behave like HSP, given its poor transcription activity compared to LSP, as reported by Tan et al. (2022).

Comment 3-3:- One aspect that I was missing in the discussion of the potential relation between the observed bending behaviour and transcriptional activity was the role of POLRMT and TFB2M. Is the authors rightly state, interactions between POLRMT and the promoter DNA are involved in initiation complex formation. Could it be that these interactions contribute substantially to the transcriptional activity, especially at HSP where the F state is less stable?

Response: Regarding the comment about POLRMT recognizing LSP versus HSP differentially, Uchida et al. (2017) reported that the basal activity of POLRMT and TFB2M (without TFAM) is higher on HSP than on LSP (see the excerpted Figure 1d from the Uchida 2017 paper above in the response to *Comment 3-2*). This may indicate that the POLRMT-HSP initiation complex may be more stable than the POLRMT-LSP complex.

We have added this information at the end of the last paragraph of Discussion as follows (see the boldfaced sentences).

Original (OL472-475)	Last but not least, ... between HSP-TFAM and NS-TFAM complexes.
Revised (RL494-512)	Last but not least, ... between HSP-TFAM and NS-TFAM complexes. A study ¹⁴ showed..., which is not the case for NS. It was also reported that the basal transcription activity of POLRMT and TFB2M (without TFAM) is higher on HSP than on LSP¹⁴, which may play an important complementary role in stabilizing the transcription initiation complex on HSP despite the less stable F-state in HSP-TFAM. Supporting this idea, our unpublished ensemble FRET data indeed showed that the addition of POLRMT and TFB2M compensated for the reduced bending ability of the C-tail mutant TFAM for both LSP and HSP, resulting in increased changes in FRET values. Systematic smFRET studies on DNA-TFAM bending dynamics in the presence of POLRMT and/or TFB2M will provide further clues to the puzzle related to HSP.

Comment 3-4: What effect does addition of POLRMT and TFB2M have on the kinetics and stability of P and F state formation?

Response: It will be interesting to investigate the effect of POLRMT and TFB2M on the DNA bending dynamics of TFAM on LSP versus HSP. However, the methods to study the three proteins in a complex have not been established yet.

Nevertheless, our unpublished ensemble FRET experiments results provide valuable insights into what the impact of POLRMT and TFB2M addition could be on the kinetics and stability of P and F state formation (see the attached figure below). Using the C-tail mutant TFAM protein ($\Delta 26$), as demonstrated in the current manuscript, to only adopt a partially bent state, ensemble FRET values were lower compared to wild-type TFAM. Notably, the addition of POLRMT and TFB2M compensated for the reduced bending ability of the C-tail mutant TFAM, resulting in increased changes in FRET values. These findings suggest that the addition of POLRMT and

Ensemble FRET data showing the effect of POLRMT and TFB2M on DNA bending. This figure is made from our unpublished ensemble FRET data on DNA bending.

TFB2M contributes to stabilizing the initiation complex, in which TFAM bends DNA in a fully bent state. Interestingly, Uchida et al. (2017) showed that the C-tail mutant TFAM activates POLRMT on LSP (but not on HSP) although the transcription product yield was 3-fold lower than WT TFAM. All these information indicate the complementary role of POLRMT and TFB2M to TFAM in bending DNA to initiate transcription.

These are unpublished data and, hence, not discussed in this manuscript, but the comment made is now discussed in the revised manuscript at the end of the last paragraph of Discussion as follows (see the boldfaced sentences).

Original (OL472-475)	Last but not least, ... between HSP-TFAM and NS-TFAM complexes.
Revised (RL494-512)	Last but not least, ... between HSP-TFAM and NS-TFAM complexes. A study ¹⁴ showed..., which is not the case for NS. It was also reported that the basal transcription activity of POLRMT and TFB2M (without TFAM) is higher on HSP than on LSP ¹⁴ , which may play an important complementary role in stabilizing the transcription initiation complex on HSP despite the less stable F-state in HSP-TFAM. Supporting this idea, our unpublished ensemble FRET data indeed showed that the addition of POLRMT and TFB2M compensated for the reduced bending ability of the C-tail mutant TFAM for both LSP and HSP, resulting in increased changes in FRET values. This complementary role of

	POLRMT and TFB2M in bending DNA was shown to rescue transcription activity even with the C-tail mutant TFAM¹⁴. Systematic smFRET studies on DNA-TFAM bending dynamics in the presence of POLRMT and/or TFB2M will provide further clues to the puzzle related to HSP.
--	---

Reviewers' Comments:

Reviewer #1:

Remarks to the Author:

After the first revision by the authors, the manuscript improved substantially.

The authors answered my comments point by point. I would suggest moving the definition of smPIFE and SMFRET to the results section, wherever appropriate (my comment was confusing I indicated line 73-74, but these were comments to the Results).

Reviewer #2:

Remarks to the Author:

The authors have addressed all of the concerns raised by the reviewers.

Reviewer #3:

Remarks to the Author:

The authors have adequately addressed all my points.

AUTHOR RESPONSES TO REVIEWER COMMENTS (on the first revision)

Note: To clearly indicate the line numbers where texts are modified, we have indicated both the line numbers corresponding to the original and the revised manuscripts with 'OL' and 'RL' prefixes, respectively: OL123-125 and RL132-138 for example.

Reviewer #1 (Remarks to the Author):

Comment: *After the first revision by the authors, the manuscript improved substantially. The authors answered my comments point by point. I would suggest moving the definition of smPIFE and SMFRET to the results section, wherever appropriate (my comment was confusing I indicated line 73-74, but these were comments to the Results).*

Response: We thank the reviewer for assessing our revised manuscript positively. We have made the requested changes as follows:

Original (L72-81)	In this work, we studied the real-time dynamics of TFAM-induced DNA bending and unbending conformational changes at the single-molecule level using smFRET and single-molecule protein-induced fluorescence enhancement (smPIFE) assays, in combination with ensemble measurements. In smFRET assay, ratiometric measurement of fluorescence from donor and acceptor that are attached to the ends of a DNA reports the real-time change in the end-to-end distance accompanied by DNA bending: the larger the bend angle, the shorter the end-to-end distance, resulting in brighter acceptor fluorescence and thus higher FRET value. In contrast, smPIFE allows to monitor the proximity of a protein to a specific position on DNA where a PIFE-compatible dye (e.g., cyanine dye Cy3) is attached and reports the protein-DNA distance by the fluorescence intensity. We found that ...
Revised (RL72-75)	In this work, we study the real-time dynamics of TFAM-induced DNA bending and unbending conformational changes at the single-molecule level using smFRET and single-molecule protein-induced fluorescence enhancement (smPIFE) assays, in combination with ensemble measurements. We find that ...
Original (L97-99)	To monitor DNA conformational dynamics with smFRET, a 30bp LSP sequence was end-labeled with Atto565 (donor) and Atto647N (acceptor) at -41 and -12 positions, respectively, and with (PEG-18) ₂ -biotin at the -41 end for surface immobilization (Fig. 1A, B). The labeled DNA was ...
Revised (RL91-97)	To monitor DNA conformational dynamics with smFRET, a 30bp LSP sequence was end-labeled with Atto565 (donor) and Atto647N (acceptor) at -41 and -12 positions, respectively, and with (PEG-18) ₂ -biotin at the -41 end for surface immobilization (Fig. 1A, B). In the smFRET assay, ratiometric measurement of fluorescence from donor and acceptor that are attached to the ends of a DNA reports the real-time change in the end-to-end distance accompanied by DNA bending: the larger the bend angle, the shorter the end-to-end distance, resulting in brighter acceptor fluorescence and thus higher FRET value. The labeled DNA was sparsely attached ...
Original (L151-155)	To distinguish between the two mechanisms of DNA bending/unbending (i.e., Model 1 vs. Model 2), we used ensemble and single-molecule protein-induced fluorescence enhancement (smPIFE) experiments. The fluorescence intensities of Cy3 or Cy5 fluorophores depend on the local environment, which influences the cis-trans isomerization dynamics of cyanine dyes ^{23,24} . To test Model 1 and Model 2A, ...

Revised (RL149-154)	To distinguish between the two mechanisms of DNA bending/unbending (i.e., Model 1 vs. Model 2), we used ensemble and single-molecule protein-induced fluorescence enhancement (smPIFE) experiments. smPIFE allows to monitor change in the proximity of a protein to a specific position on DNA where a PIFE-compatible dye (e.g., cyanine dye Cy3) is attached; because change in the local environment influences the cis-trans isomerization dynamics of cyanine dyes ^{23,24} and thus the fluorescence intensity. To test Model 1 and Model 2A, ...
--

Reviewer #2 (Remarks to the Author):

Comment: *The authors have addressed all of the concerns raised by the reviewers.*

Response: We thank the reviewer for assessing our revised manuscript positively. No further action is required.

Reviewer #3 (Remarks to the Author):

Comment: *The authors have adequately addressed all my points.*

Response: We thank the reviewer for assessing our revised manuscript positively. No further action is required.